# Beyond Global Alignment: Fine-Grained Motion-Language Retrieval via Pyramidal Shapley-Taylor Learning

Hanmo Chen [1]   Guangtao Lyu [2]   Chenghao Xu [3]   Jiexi Yan [2]   Xu Yang [2]   Cheng Deng [2]

## Abstract

As a foundational task in human-centric cross-modal intelligence, motion-language retrieval aims to bridge the semantic gap between natural language and human motion, enabling intuitive motion analysis, yet existing approaches predominantly focus on aligning entire motion sequences with global textual representations. This global-centric paradigm overlooks fine-grained interactions between local motion segments and individual body joints and text tokens, inevitably leading to suboptimal retrieval performance. To address this limitation, we draw inspiration from the pyramidal process of human motion perception (from joint dynamics to segment coherence, and finally to holistic comprehension) and propose a novel Pyramidal Shapley-Taylor (PST) learning framework for fine-grained motion-language retrieval. Specifically, the framework decomposes human motion into temporal segments and spatial body joints, and learns cross-modal correspondences through progressive joint-wise and segment-wise alignment in a pyramidal fashion, effectively capturing both local semantic details and hierarchical structural relationships. Extensive experiments on multiple public benchmark datasets demonstrate that our approach significantly outperforms state-of-the-art methods, achieving precise alignment between motion segments and body joints and their corresponding text tokens.

## 1. Introduction

In recent years, motion-language retrieval (Yu et al., 2024) has garnered increasing attention in the field of human-centric cross-modal intelligence (Xu et al., 2026; Li et al., 2018; Chen et al.), as it holds significant potential to bridge the semantic gap between natural language and human motion (Li et al., 2025; Chen et al., 2025a). As a core foundational research topic, motion-language retrieval underpins a diverse range of downstream tasks, such as human motion understanding (Lyu et al., 2025; Schneider et al., 2024; Fang et al., 2025; Chen et al., 2024), motion style transfer (Chen et al., 2025b; Kim et al., 2025; Zhong et al., 2024), and motion generation (Meng et al., 2025; Hong et al., 2025; Chen et al., 2023; Wang et al., 2025).

Despite substantial advancements in this field, a persistent and significant semantic gap persists between human motion representations and natural language descriptions. A critical limitation of existing approaches (Petrovich et al., 2023; Yu et al., 2024; Tevet et al., 2022a) is their predominant focus on global alignment between entire motion sequences and their corresponding textual descriptions. This global-centric paradigm overemphasizes holistic correlation modeling while failing to capture fine-grained local associations. This oversight hinders the precise matching of semantic details between the two modalities. Consequently, addressing this fine-grained alignment challenge becomes imperative for advancing motion-language retrieval performance.

Notably, human perception (Giese & Poggio, 2003) and understanding of motion (Zhong et al., 2023; Zhu et al., 2023; Xu et al., 2024) follow an inherent pyramidal process, progressing from low-level joint dynamics to high-level holistic sequence comprehension. Specifically, this cognitive process unfolds in three sequential stages: First, at the joint-wise alignment stage, humans perceive the temporal trajectories of individual joints and interpret relative movements between joints, laying the foundation for recognizing short motion segments. Second, at the segment-wise alignment stage, by integrating these individual joints into meaningful motion segments, dynamic relations are established between adjacent motion segments. Finally, at the holistic alignment stage, these motion segments are synthesized into a unified, holistic model, in which localized kinematic features and semantic information are hierarchically integrated to achieve comprehensive motion understanding.

To illustrate this process, consider the textual description "a

[1] Hangzhou Institute of Technology, Xidian University, Hangzhou, China [2] Xidian University, Xi'an, China [3] Hohai University, Nanjing, China. Correspondence to: Xu Yang <xuyang.xd@gmail.com>.

*Proceedings of the $43^{rd}$ International Conference on Machine Learning*, Seoul, South Korea. PMLR 306, 2026. Copyright 2026 by the author(s).

man walks forward then stops". At the joint-wise alignment stage, we first detect periodic swings of the arms and legs, forward translation of the root joint, and the final stationary pose. During segment-wise alignment, we identify coordinated motion patterns corresponding to "walks forward" and "stops" by analyzing the temporal synchronization between limb movements and root joint trajectories. Ultimately, integrating these two sub-sequences enables holistic comprehension of the entire motion described. Inspired by this human cognitive mechanism, we introduce this pyramidal hierarchical framework into motion-language retrieval, empowering the model to progressively capture cross-modal correspondences from fine-grained local details to holistic semantic structures.

Given that human motion inherently exhibits temporal continuity and strong contextual dependencies between adjacent motion segments as well as body joints, capturing such context-aware characteristics is critical for modeling accurate motion-language correspondence. Inspired by Jin et al. (2023); Wang et al. (2025), we introduce a Shapley-Taylor Interaction (Sundararajan et al., 2020) (STI) metric which quantifies the interaction intensity between pairs of cross-modal elements across varying contextual scenarios, to address this requirement. Specifically, as contextual information is gradually incorporated, Shapley-Taylor Interaction measures the additional marginal contribution achieved when two elements co-occur versus when they appear independently, with higher STI values indicating stronger semantic correlations between the corresponding motion and language elements. Building on this, we propose a Pyramidal Shapley-Taylor (PST) learning framework, which embeds STI within the pyramidal modeling scheme to capture joint-wise and segment-wise motion-language interactions.

The proposed PST framework adopts a progressive pyramidal training strategy consisting of three hierarchical levels: i) Joint-wise alignment, which calculates interaction strengths between individual word tokens and joint tokens; ii) Segment-wise alignment, which compresses joint tokens and word tokens into segment-level and phrase-level tokens via dedicated token compressors, then computes their cross-modal interactions; iii) Holistic alignment, which treats all motion and text tokens as unified global representations and quantifies their global-level correspondence. Through this PST learning framework, the model effectively captures motion-language correspondences at fine-grained scales, facilitating more precise motion-language retrieval. Furthermore, we present joint-wise and segment-wise visualization analyses that explicitly demonstrate how the model aligns motion units with textual descriptions. Extensive experiments on public benchmark datasets verify that our approach attains state-of-the-art retrieval performance. Our contributions can be summarized as follows:

- We propose a Pyramidal Shapley-Taylor (PST) learning framework that performs fine-grained motion-language retrieval in a pyramidal fashion via joint-wise, segment-wise, and holistic alignment.

- The visualization results across joint-wise and segment-wise alignments demonstrate the effectiveness of our method in capturing fine-grained motion-language correspondences.

- Extensive experiments across multiple public datasets show that our model achieves state-of-the-art results with remarkable effectiveness and generalization across diverse motion-language tasks.

## 2. Related Work

**Motion-Language Learning.** Motion-language learning has gained substantial attention for narrowing the gap between human motion and natural language. Petrovich et al. (2023) extends TEMOS (Petrovich et al., 2022) with contrastive learning and introduces TMR, achieving a more structured cross-modal latent space for text-motion retrieval. Yu et al. (2024) introduces MotionPatch, which slices motion sequences by body parts and encodes them with pretrained Vision Transformer (ViT), leading to effective motion-language retrieval. However, these approaches process motion and text as holistic inputs, which departs from a fine-grained human perspective when describing motion sequences. To move toward interpretable understanding, Lyu et al. (2025) introduce lexicalized (Shen et al., 2022) motion representations that render motion-language features more transparent and controllable. Chronologically Accurate Retrieval (CAR), proposed by Fujiwara et al. (2024), decomposes textual descriptions into several temporal elements to improve temporal correspondence between language and motion.

**Motion-Language Retrieval for Generation.** Motion-language learning has been widely applied in human motion generation tasks. ReMoDiffuse, proposed by Zhang et al. (2023b), enhances the generalizability and diversity of motion diffusion models through an integrated motion-language retrieval mechanism. Similarly, Yu et al. (2025) achieves body-part-level motion retrieval to improve the controllability of generated motions. Moreover, MoRAG, introduced by Kalakonda et al. (2025), leverages Large Language Models (LLMs) to mitigate spelling errors and rephrasing issues during motion retrieval, thereby improving the robustness and generalization of motion generation. These studies collectively demonstrate that motion-language retrieval plays a crucial role in advancing motion generation and reveal that developing fine-grained motion-language retrieval is essential for facilitating controllable and semantically consistent motion generation.

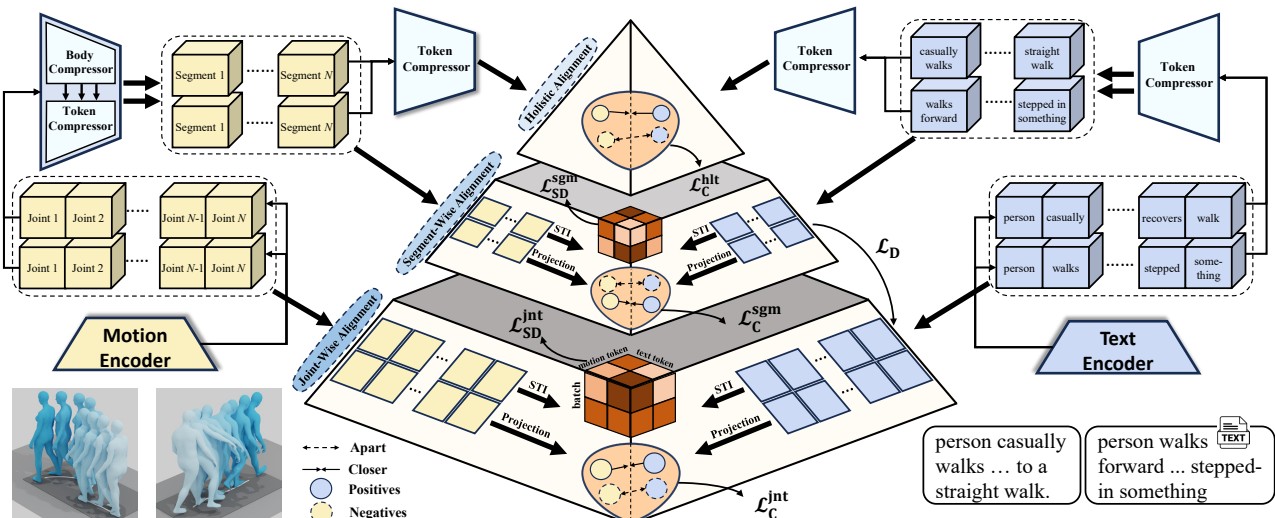

*Figure 1.* Overview of our Pyramidal Shapley-Taylor (PST) learning framework. Our PST learning framework consists of Shapley-Taylor Interaction (STI), described in Sec. 3.2, and pyramidal modeling scheme, described in Sec. 3.3. As illustrated in the middle cube, each cell represents the interaction strength between a motion token and a text token within a batch, where darker colors indicate stronger semantic correlations, and lighter colors represent weaker ones.

## 3. Method

### 3.1. Preliminary

Motion-language retrieval can be formulated as a cross-modal alignment problem (Deng et al., 2018; Yang et al., 2019; Xu et al., 2025), whose objective is to establish a structured cross-modal latent space between the set of text descriptions $\mathcal{T} = \{t_i\}_{i=1}^{N_t}$ and the set of human motion sequences $\mathcal{M} = \{m_j\}_{j=1}^{N_m}$, where $N_t$ and $N_m$ denote the number of text descriptions and motion sequences, respectively. Specifically, given a text description $t_i \in \mathcal{T}$, the goal is to retrieve and rank the motion sequences that best match the given text description from a motion database. Conversely, given a motion sequence $m_j \in \mathcal{M}$, the goal is also to retrieve the most relevant text description that best matches its motion content. A text description $t_i$ provides a natural-language explanation of either the entire motion sequence or specific temporal segments within it (Zhang et al., 2023a; Sun et al., 2024; Cen et al., 2024), e.g., "Walk forward and stand still" and "Walk forward". Moreover, the text may contain references to specific body parts or global motion directions, such as "the left arm waves" or "walk in a circle". The motion sequence $m_j$ is a discrete temporal representation (Jang et al., 2022; Tevet et al., 2022b; Punnakkal et al., 2021) of human motion. It is defined as $m_j \in \mathbb{R}^{L \times J \times C}$, where $L$, $J$, and $C$ denote the length of frames, number of joints, and the dimension of the input motion, respectively. In this paper, we use 3D skeleton joints as the motion representation, where $C = 3$ denotes the Cartesian coordinates $(x, y, z)$ of each joint.

To achieve motion-language retrieval, we first employ a

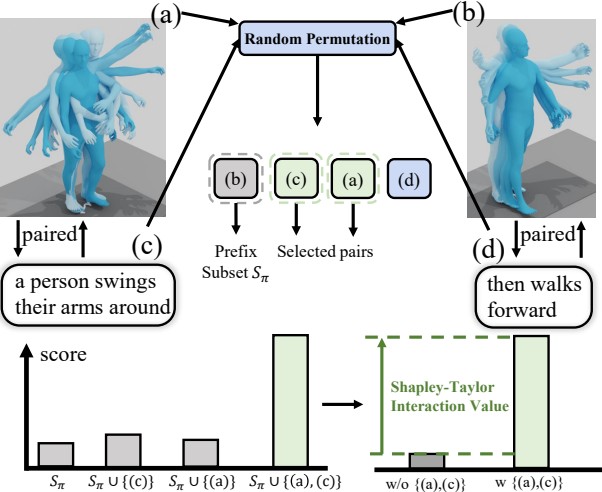

*Figure 2.* An intuitive illustration of the STI.

modality-specific encoder $\mathcal{E}$ to project $t_i$ and $m_j$ into a latent space, and then use the cosine similarity function to measure the similarity between them. The cosine similarity function is formulated as:

$$s(t_i, m_j) = \frac{\mathcal{E}_T(t_i) \cdot \mathcal{E}_M(m_j)}{\|\mathcal{E}_T(t_i)\| \|\mathcal{E}_M(m_j)\|}, \tag{1}$$

where $\mathcal{E}_M$ and $\mathcal{E}_T$ denote motion encoder and text encoder, respectively, as used in our framework.

## 3.2. Shapley-Taylor Interaction

The Shapley-Taylor Interaction (STI) (Sundararajan et al., 2020) extends the Shapley Interaction by taking a Taylor expansion truncated at order $k$, where $k = 1$ recovers standard Shapley attributions and $k > 1$ quantifies interaction among cross-modality features. In our motion-language retrieval task, which analyzes pairwise interactions between motion and language modalities, we incorporate STI into our learning framework and set $k = 2$. Given a set of text tokens and motion tokens, the STI first concatenates the two modalities into a unified set and performs a random permutation over all tokens. For a selected text token $e_i^{\mathrm{t}}$ and motion token $e_j^{\mathrm{m}}$, under a random permutation $\pi$, let $\mathrm{pos}_\pi(\cdot)$ denote the position of a token in the permutation. The prefix subset $S_\pi$ is then composed of all tokens appearing before $\min(\mathrm{pos}_\pi(e_i^{\mathrm{t}}), \mathrm{pos}_\pi(e_j^{\mathrm{m}}))$. On this $S_\pi$, we compute STI value $\phi$ of the token pair $(e_i^{\mathrm{t}}, e_j^{\mathrm{m}})$ as follows:

$$
\begin{aligned}
\phi(e_i^{\mathrm{t}}, e_j^{\mathrm{m}}) = \mathbb{E}_\pi \Big[ & F(S_\pi \cup \{e_i^{\mathrm{t}}, e_j^{\mathrm{m}}\}) - F(S_\pi \cup \{e_i^{\mathrm{t}}\}) \\
& - F(S_\pi \cup \{e_j^{\mathrm{m}}\}) + F(S_\pi) \Big],
\end{aligned} \tag{2}
$$

where $F(\cdot)$ denotes the scoring function, which is defined by the similarity score in our retrieval task, and $\mathbb{E}_\pi[\cdot]$ denotes the expectation taken over all random permutations. Fig. 2 provides an intuitive understanding of the STI. Conceptually, under each permutation $\pi$, the model first takes $S_\pi$ as the input to compute scoring function, then compares the score changes when $e_i^{\mathrm{t}}$, $e_j^{\mathrm{m}}$, or both are added into $S_\pi$. The change in score reflects the marginal contribution of the token pair $(e_i^{\mathrm{t}}, e_j^{\mathrm{m}})$ under this permutation, and averaging these scores over all permutations yields the final STI value for this pair.

Since STI explicitly relies on random permutations and their corresponding prefix subsets, it can be interpreted as follows: starting from an empty subset, the model gradually adds tokens into this subset. When the token pair $(e_i^{\mathrm{t}}, e_j^{\mathrm{m}})$ is inserted, its STI value measures the additional gain in the final retrieval score contributed by this pair conditioned on the current prefix subset. Formally, consider a token set of size $K$, we define variable $L$ to denote the length of prefix subset. The probability distribution $\mathbb{P}$ of $L$ is given by:

$$
\mathbb{P}(L = \ell) = \frac{2(K - \ell - 1)}{K(K-1)}, \ell = 0, 1, \ldots, K-2, \tag{3}
$$

where $K \geq 2$. For the detailed explanation of Equation 3, please refer to the Appendix. This distribution indicates that shorter prefixes, where $\ell$ is small, occur more frequently, while longer prefixes have smaller probabilities. Consequently, smaller subsets $S_\pi$ are assigned larger weights in the overall expectation, whereas larger subsets contribute less. From a semantic perspective, when $S_\pi$ is small, the available contextual information is limited, leading to substantial improvement in the retrieval score when adding the

pair $(e_i^{\mathrm{t}}, e_j^{\mathrm{m}})$. Conversely, when $S_\pi$ is large, containing other well-aligned pairs, $F(S_\pi)$ may have reached near-saturation, causing the marginal contribution gain from adding the paired $(e_i^{\mathrm{t}}, e_j^{\mathrm{m}})$ to diminish. This property causes STI values to exhibit distinctive peaks for genuinely aligned motion-language pairs, effectively highlighting the most critical motion-language correspondences.

However, due to the calculation across all possible permutations, the computational cost of STI is prohibitively high during the training stage. To address this issue, we construct a lightweight approximation model, named the STI Estimation Head $\mathcal{H}$, composed of convolution and self-attention (Vaswani et al., 2017) layers, and train it using Monte Carlo sampling of STI. Concretely, to integrate this $\mathcal{H}$ into training, we transform STI values into token-wise probability distributions of motion-to-text and text-to-motion STI values, formulated as follows:

$$
\begin{aligned}
p_{i,j}^\phi &= \frac{\exp(\phi(e_i^{\mathrm{t}}, e_j^{\mathrm{m}}))}{\sum_{k=1}^{N_{\mathrm{t}}} \exp(\phi(e_k^{\mathrm{t}}, e_j^{\mathrm{m}}))}, \\
\hat{p}_{i,j}^\phi &= \frac{\exp(\phi(e_i^{\mathrm{t}}, e_j^{\mathrm{m}}))}{\sum_{k=1}^{N_{\mathrm{m}}} \exp(\phi(e_i^{\mathrm{t}}, e_k^{\mathrm{m}}))}.
\end{aligned} \tag{4}
$$

For each motion token $e_j^{\mathrm{m}}$ and each text token $e_i^{\mathrm{t}}$, we obtain the motion-to-text distribution and the text-to-motion distribution respectively, formulated as follows:

$$
\begin{aligned}
\mathcal{D}_{\mathrm{m2t}}^\phi(j) &= [p_{1,j}^\phi, p_{2,j}^\phi, \ldots, p_{N_{\mathrm{t}},j}^\phi], \\
\mathcal{D}_{\mathrm{t2m}}^\phi(i) &= [\hat{p}_{i,1}^\phi, \hat{p}_{i,2}^\phi, \ldots, \hat{p}_{i,N_{\mathrm{m}}}^\phi].
\end{aligned} \tag{5}
$$

Similarly to Equation 4, the STI Estimation Head $\mathcal{H}$ takes a motion token $m_j$ and a text token $t_i$ as input and produces the estimated STI value $p_{i,j}^\mathcal{H}$ and $\hat{p}_{i,j}^\mathcal{H}$, which form the motion-to-text and text-to-motion distributions $\mathcal{D}_{\mathrm{m2t}}^\mathcal{H}$ and $\mathcal{D}_{\mathrm{t2m}}^\mathcal{H}$ analogous to Equation 5. These distributions serve as student predictions, trained to match the standard STI distributions via our STI distillation loss $\mathcal{L}_{\mathrm{SD}}$, defined using a Kullback-Leibler (KL) divergence (Kingma & Welling, 2013; Kullback, 1997) objective. Our $\mathcal{L}_{\mathrm{SD}}$ is formulated as follows:

$$
\mathcal{L}_{\mathrm{SD}} = \mathrm{KL}\left(\mathcal{D}_{\mathrm{m2t}}^\phi \| \mathcal{D}_{\mathrm{m2t}}^\mathcal{H}\right) + \mathrm{KL}\left(\mathcal{D}_{\mathrm{t2m}}^\phi \| \mathcal{D}_{\mathrm{t2m}}^\mathcal{H}\right). \tag{6}
$$

This lightweight model learns to approximate the STI without enumerating all permutations, enabling efficient integration of STI into our motion-language retrieval task while maintaining its interpretability and fine-grained interaction modeling capabilities.

## 3.3. Pyramidal Modeling Scheme

Traditional retrieval methods often treat text and motion sequences as a whole, failing to capture fine-grained cross-modal feature interactions. This limitation motivates us to introduce a pyramidal modeling scheme that enables the model to capture detailed motion-language correspondences at multiple scales progressively. We define joint-wise alignment, segment-wise alignment, and holistic alignment as key stages in the retrieval process.

To enhance the ability of the model to capture contextual dependencies within motion sequences, we leverage a token compression mechanism between adjacent alignment stages. Our token compressor consists of convolution, self-attention (Vaswani et al., 2017), and a K-Nearest Neighbor-based Density Peaks Clustering (KNN-DPC) (Du et al., 2016) algorithm, which selects cluster centers based on high local density and significant relative distance, and assigns each remaining token to its nearest center.

In joint-wise alignment, the model performs fine-grained retrieval between each motion joint and its corresponding textual tokens over time. Given a text description $t$ and a motion sequence $m$ as input, we encode them using a text encoder $\mathcal{E}_T$ and a motion encoder $\mathcal{E}_M$, respectively. To preserve the structural properties of the motion sequence, we encode the motion along the joint dimension $d_j$ and the temporal $d_t$ dimension, and then flatten them to obtain $N_{\mathrm{m}}^{\mathrm{jnt}} = d_j \times d_t$. This yields text and motion embeddings $E^{*,(\mathrm{jnt})} = \{e_i^{*,(\mathrm{jnt})}\}_{i=1}^{N_*^{\mathrm{jnt}}} \in \mathbb{R}^{N_*^{\mathrm{jnt}} \times D}$, where $* \in [\mathrm{m}, \mathrm{t}], e_i^{*,(\mathrm{jnt})}$ and $N_*^{\mathrm{jnt}}$ denote the joint-stage token and the number of token, respectively. In segment-wise alignment, joint-stage text embedding $E^{\mathrm{t},(\mathrm{jnt})}$ is passed through our token compressor to produce segment-stage text embedding $E^{\mathrm{t},(\mathrm{sgm})} = \{e_i^{\mathrm{t},(\mathrm{sgm})}\}_{i=1}^{N_{\mathrm{t}}^{\mathrm{sgm}}} \in \mathbb{R}^{N_{\mathrm{t}}^{\mathrm{sgm}} \times D}$ with text compression ratio of $\rho_{\mathrm{t}}$. For the motion modality, we first reshape $E^{\mathrm{m},(\mathrm{jnt})}$ to $\mathbb{R}^{d_j \times d_t \times D}$ and apply a body compressor to compress along the whole joint axis, and subsequently apply the token compressor along the temporal dimension with a motion compression ratio of $\rho_{\mathrm{m}}$, yielding segment-stage motion embedding $E^{\mathrm{m},(\mathrm{sgm})} = \{e_j^{\mathrm{m},(\mathrm{sgm})}\}_{j=1}^{N_{\mathrm{m}}^{\mathrm{sgm}}} \in \mathbb{R}^{N_{\mathrm{m}}^{\mathrm{sgm}} \times D}$. The compression ratio, defined as $\rho_* = N_*^{\mathrm{sgm}}/N_*^{\mathrm{jnt}}$, where $* \in \{\mathrm{m}, \mathrm{t}\}$, is set to 0.25 in our experiments, and a detailed ablation study on the influence of $\rho_*$ is presented in Sec. 4.3. In holistic alignment, the segment-stage text embeddings $E^{\mathrm{t},(\mathrm{sgm})}$ and motion embeddings $E^{\mathrm{m},(\mathrm{sgm})}$ are each aggregated into a holistic representation, yielding text and motion embeddings $E^{\mathrm{t},(\mathrm{hlt})}, E^{\mathrm{m},(\mathrm{hlt})} \in \mathbb{R}^D$.

## 3.4. Training Objective

The training objective of PST framework is defined through the contrastive loss $\mathcal{L}_{\mathrm{C}}$ for each specific alignment stage

mentioned in Sec. 3.3 and the STI distillation loss $\mathcal{L}_{\mathrm{SD}}$ described in Sec. 3.2 for the joint-wise and segment-wise alignment stage. Since the text and motion embeddings $E^{\mathrm{t}}$, $E^{\mathrm{m}}$ are token-wise representations, we first use a projection head $PH(\cdot)$ to remove the feature dimension and modify the similarity function defined in Equation 1:

$$s(t_i, m_j) = \frac{PH(\mathcal{E}_T(t_i)) \cdot PH(\mathcal{E}_M(m_j))}{\|PH(\mathcal{E}_T(t_i))\|\|PH(\mathcal{E}_M(m_j))\|}. \quad (7)$$

Given a batch size of $B$, the contrastive loss $\mathcal{L}_{\mathrm{C}}$ is defined as Information Noise-Contrastive Estimation (InfoNCE) loss and formulated as follows:

$$\mathcal{L}_{\mathrm{t2m}} = -\frac{1}{B} \sum_{i=1}^{B} \log \frac{\exp(s(t_i, m_i)/\tau)}{\sum_{j=1}^{B} \exp(s(t_i, m_j)/\tau)},$$

$$\mathcal{L}_{\mathrm{m2t}} = -\frac{1}{B} \sum_{i=1}^{B} \log \frac{\exp(s(t_i, m_i)/\tau)}{\sum_{j=1}^{B} \exp(s(t_j, m_i)/\tau)}, \quad (8)$$

$$\mathcal{L}_{\mathrm{C}} = \mathcal{L}_{\mathrm{t2m}} + \mathcal{L}_{\mathrm{m2t}},$$

where $\tau$ denotes the temperature hyperparameter. Although our PST learning framework enables joint-wise, segment-wise, and holistic alignment, we observe an inherent inconsistency during training: the similarity distributions across different semantic levels do not converge synchronously, leading to suboptimal alignment quality. We attribute this issue to the differences in similarity distributions across alignment stages. To address this, we introduce a self-distillation loss based on KL divergence, where the similarity distributions at the joint-wise alignment stage are used as teacher signals to guide the similarities at the segment-wise alignment stage. Specifically, we compute similarity distributions for motion-text and text-motion pairs at joint-wise and segment-wise alignment stages, and encourage the model to preserve the semantic consistency learned at the joint-wise alignment stage when training at the segment-wise alignment stage through KL loss, defined as follows:

$$\mathcal{L}_{\mathrm{D}} = \mathrm{KL}(\mathcal{D}_{\mathrm{m2t}}^{\mathrm{sgm}} \| \mathcal{D}_{\mathrm{m2t}}^{\mathrm{jnt}}) + \mathrm{KL}(\mathcal{D}_{\mathrm{t2m}}^{\mathrm{sgm}} \| \mathcal{D}_{\mathrm{t2m}}^{\mathrm{jnt}}). \quad (9)$$

Consequently, the overall loss function of our PST learning framework is formulated as follows:

$$\mathcal{L} = \mathcal{L}_{\mathrm{C}}^{\mathrm{jnt}} + \mathcal{L}_{\mathrm{C}}^{\mathrm{sgm}} + \mathcal{L}_{\mathrm{C}}^{\mathrm{hlt}} + \lambda_{\mathrm{S}}(\mathcal{L}_{\mathrm{SD}}^{\mathrm{jnt}} + \mathcal{L}_{\mathrm{SD}}^{\mathrm{sgm}}) + \lambda_{\mathrm{D}}\mathcal{L}_{\mathrm{D}}, \quad (10)$$

where $\lambda_*$ denotes the weight of each loss. The losses $\mathcal{L}_{\mathrm{C}}^{\mathrm{jnt}}$, $\mathcal{L}_{\mathrm{C}}^{\mathrm{sgm}}$ and $\mathcal{L}_{\mathrm{C}}^{\mathrm{hlt}}$ represent the contrastive loss in joint-wise, segment-wise and holistic alignment stages, respectively, while $\mathcal{L}_{\mathrm{SD}}^{\mathrm{jnt}}$ and $\mathcal{L}_{\mathrm{SD}}^{\mathrm{sgm}}$ denote the STI distillation loss at the joint-wise and segment-wise alignment stages, respectively. We set the hyperparameters $\lambda_{\mathrm{S}} = 1$ and $\lambda_{\mathrm{D}} = 1$ in experiments, and provide a detailed ablation study in Sec. 4.3.

*Table 1.* Motion-to-text and text-to-motion retrieval results on HumanML3D.

| Protocol | Methods | Text-to-motion retrieval | | | | | | Motion-to-text retrieval | | | | | |
|---|---|---|---|---|---|---|---|---|---|---|---|---|---|
| | | R@1↑ | R@2↑ | R@3↑ | R@5↑ | R@10↑ | MedR↓ | R@1↑ | R@2↑ | R@3↑ | R@5↑ | R@10↑ | MedR↓ |
| All | TEMOS(Petrovich et al., 2022) | 2.12 | 4.09 | 5.87 | 8.26 | 13.52 | 173 | 3.86 | 4.54 | 6.94 | 9.38 | 14.00 | 183.25 |
| | T2M(Guo et al., 2022) | 1.80 | 3.42 | 4.79 | 7.12 | 12.47 | 81.00 | 2.92 | 3.74 | 6.00 | 8.36 | 12.95 | 81.50 |
| | TMR(Petrovich et al., 2023) | 8.92 | 12.04 | 16.33 | 22.06 | 33.37 | 25.00 | 9.44 | 11.84 | 16.90 | 22.92 | 32.21 | 26.00 |
| | MotionPatch(Yu et al., 2024) | 10.80 | 14.98 | 20.00 | 26.72 | 38.02 | 19.00 | 11.25 | 13.86 | 19.98 | 26.86 | 37.40 | 20.50 |
| | Lyu et al. (2025) | 11.80 | 17.11 | 23.25 | 30.81 | 43.36 | 14.00 | 12.39 | 15.55 | 22.17 | 29.25 | 40.34 | 17.00 |
| | Ours | **12.45** | **19.02** | **26.50** | **33.65** | **48.22** | **10.00** | **13.59** | **17.11** | **24.42** | **31.88** | **42.68** | **15.00** |
| Small batches | TEMOS(Petrovich et al., 2022) | 40.49 | 53.52 | 61.14 | 70.96 | 84.15 | 2.33 | 39.96 | 53.49 | 61.79 | 72.40 | 85.89 | 2.33 |
| | T2M(Guo et al., 2022) | 52.48 | 71.05 | 80.65 | 89.66 | 96.58 | 1.39 | 52.00 | 71.21 | 81.11 | 89.87 | 96.78 | 1.38 |
| | TMR(Petrovich et al., 2023) | 67.45 | 80.98 | 86.22 | 91.56 | 95.46 | 1.03 | 68.59 | 81.73 | 86.75 | 91.10 | 95.39 | 1.02 |
| | MotionPatch(Yu et al., 2024) | 71.61 | 85.81 | 90.02 | 94.35 | 97.69 | **1.00** | 72.11 | 85.26 | 90.21 | 94.44 | 97.76 | **1.00** |
| | Ours | **76.15** | **89.31** | **94.12** | **96.35** | **98.71** | **1.00** | **75.12** | **88.79** | **93.58** | **96.64** | **98.46** | **1.00** |

*Table 2.* Motion-to-text and text-to-motion retrieval results on KIT-ML.

| Protocol | Methods | Text-to-motion retrieval | | | | | | Motion-to-text retrieval | | | | | |
|---|---|---|---|---|---|---|---|---|---|---|---|---|---|
| | | R@1↑ | R@2↑ | R@3↑ | R@5↑ | R@10↑ | MedR↓ | R@1↑ | R@2↑ | R@3↑ | R@5↑ | R@10↑ | MedR↓ |
| All | TEMOS(Petrovich et al., 2022) | 7.11 | 13.25 | 17.59 | 24.10 | 35.66 | 24.00 | 11.69 | 15.30 | 20.12 | 26.63 | 36.39 | 26.50 |
| | T2M(Guo et al., 2022) | 3.37 | 6.99 | 10.84 | 16.87 | 27.71 | 28.00 | 4.94 | 6.51 | 10.72 | 16.14 | 25.30 | 28.50 |
| | TMR(Petrovich et al., 2023) | 10.05 | 13.87 | 20.74 | 30.03 | 44.66 | 14.00 | 11.83 | 13.74 | 22.14 | 29.39 | 38.55 | 16.00 |
| | MotionPatch(Yu et al., 2024) | 14.02 | 21.08 | 28.91 | 34.10 | 50.00 | 10.50 | 13.61 | 17.26 | 27.54 | 33.33 | 44.77 | 13.00 |
| | Lyu et al. (2025) | 15.13 | 23.74 | 31.61 | 36.81 | 54.12 | 8.00 | 15.01 | 19.47 | 30.06 | 35.63 | 47.53 | 10.50 |
| | Ours | **16.01** | **25.37** | **34.21** | **38.36** | **57.87** | **7.00** | **16.76** | **20.97** | **32.15** | **37.54** | **50.17** | **8.00** |
| Small batches | TEMOS(Petrovich et al., 2022) | 43.88 | 58.25 | 67.00 | 74.00 | 84.75 | 2.06 | 41.88 | 55.88 | 65.62 | 75.25 | 85.75 | 2.25 |
| | T2M(Guo et al., 2022) | 42.25 | 62.62 | 75.12 | 87.50 | 96.12 | 1.88 | 39.75 | 62.75 | 73.62 | 86.88 | 95.88 | 1.95 |
| | TMR(Petrovich et al., 2023) | 50.00 | 64.14 | 78.02 | 87.97 | 94.87 | 1.50 | 51.21 | 69.53 | 78.64 | 89.00 | 95.31 | 1.50 |
| | MotionPatch(Yu et al., 2024) | 53.55 | 71.30 | 79.82 | 88.92 | 96.29 | 1.36 | 54.54 | 72.15 | 79.68 | 89.35 | 96.11 | 1.31 |
| | Ours | **56.83** | **75.69** | **81.64** | **89.84** | **97.94** | **1.28** | **57.14** | **74.53** | **80.76** | **90.15** | **97.58** | **1.26** |

## 4. Experiments

### 4.1. Experimental Settings

**Experimental Implementation.** For a fair comparison, we follow MotionPatch (Yu et al., 2024) and take text-to-motion retrieval and motion-to-text retrieval to analyze the performance of our method. We train our models on a single NVIDIA A6000 48 GB GPU. We optimize our STI Estimation Head and retrieval model separately using Adam optimizers (Kingma & Ba, 2014) with the same $\beta_1 = 0.9$ and $\beta_2 = 0.99$, and different learning rates of $1 \times 10^{-3}$ and $1 \times 10^{-4}$, respectively. We set the batch size as 128 and trained for 100 epochs. Moreover, inspired by Motion-Patch (Yu et al., 2024), which applies spatial structuring to motion inputs and captures spatio-temporal dependencies through a Vision Transformer (ViT) (Dosovitskiy, 2020), we incorporate a similar motion representation and adopt ViT as our motion encoder. Besides, DistilBERT (Sanh et al., 2019) is also adopted as our text encoder.

**Datasets.** We evaluate our PST framework on two commonly used public motion-language datasets, the HumanML3D dataset (Guo et al., 2022) and KIT Motion-Language (KIT-ML) dataset (Plappert et al., 2016). The HumanML3D dataset is constructed from the AMASS (Mahmood et al., 2019) and HumanAct12 (Guo et al., 2020) motion capture datasets. In total, the dataset contains 14,616 motion sequences and 44,970 text descriptions, comprising

23,384 motions for training, 1,460 for validation, and 4,380 for testing. The KIT-ML dataset, which primarily focuses on locomotion, is a dataset where each motion clip is paired with one to four short natural language descriptions. It is partitioned into training, validation, and test sets, consisting of 4,888, 300, and 830 motions, respectively.

**Evaluation Metrics.** To effectively demonstrate the robustness of our proposed method, we follow previous works (Petrovich et al., 2023; Yu et al., 2024) to adopt recall at several ranks (R@$k$) and Median Rank (MedR). R@$k$ measures the proportion of queries for which the correct item appears within the top-$k$ results, where a higher value indicates better performance. In our experiments, we report R@1, R@2, R@3, R@5 and R@10. Additionally, MedR, where a lower value indicates better retrieval performance, is calculated to evaluate our model.

**Evaluation Protocol** To ensure a reliable and comparable evaluation, we follow previous works (Petrovich et al., 2023; Yu et al., 2024) that modified the construction of the retrieval gallery and conduct experiments under two different evaluation protocols, namely *All* and *Small Batch*. In the *All* protocol , the entire test set is used as the retrieval gallery. However, this configuration often includes a large number of highly similar motion-language samples, which may lead to unreliable evaluation results. Therefore, the *Small Batch* protocol is also adopted. Specifically, we randomly partition the test set into multiple batches, each containing 32 sam-

ples, and compute the retrieval metrics within each batch. The final performance is then obtained by averaging the metrics across all sampled batches, thereby providing a more robust and reliable set of evaluation results.

## 4.2. Experimental Results

**Quantitative Results.** Comprehensive evaluations are conducted for both text-to-motion and motion-to-text retrieval tasks on the HumanML3D and KIT-ML datasets under all evaluation protocols. We compare our method against several approaches, including TEMOS (Petrovich et al., 2022), T2M (Guo et al., 2022), TMR (Petrovich et al., 2023), MotionPatch (Yu et al., 2024), and Lyu et al. (2025). Since TEMOS (Petrovich et al., 2022) and T2M (Guo et al., 2022) were originally designed for motion generation rather than retrieval, we adopt the retrieval metrics reported by TMR (Petrovich et al., 2023), which retrained and evaluated these models under a retrieval setting, for a fair comparison. Evaluation results on HumanML3D and KIT-ML are summarized in Table 1 and Table 2, respectively. As shown in Table 1 and Table 2, TEMOS (Petrovich et al., 2022) and T2M (Guo et al., 2022) perform poorly in the retrieval task, while TMR (Petrovich et al., 2023), which extends TEMOS (Petrovich et al., 2022) by incorporating a contrastive learning objective, achieves better retrieval performance. MotionPatch (Yu et al., 2024) further improves retrieval accuracy by capturing spatiotemporal features using a ViT (Dosovitskiy, 2020), and Lyu et al. (2025) enhance performance by integrating a lexical representation paradigm into the motion-language learning framework. Nevertheless, our method, which introduces fine-grained motion-language alignment into the retrieval process, outperforms all prior methods across both datasets, achieving state-of-the-art performance.

**Qualitative Results.** To better demonstrate the advantages of our method in fine-grained retrieval, we visualize the retrieval results at both the joint-wise and segment-wise alignment stages in Fig. 4 and Fig. 3, respectively. Since our approach adopts the motion representation from MotionPatch and employs a ViT to model both spatial and temporal dependencies, the motion tokens at the joint-wise stage effectively encode spatial as well as temporal characteristics. As a result, the examples shown in Fig. 4 exhibit strong semantic correlations between text descriptions and joint tokens. Additionally, in the segment-wise stage, we employ the KNN-DPC algorithm, which not only clusters tokens at the segment-level but also preserves token provenance information, to perform token compression, allowing us to trace back which joint tokens are aggregated into each compressed segment token. Segment-wise alignment results in Fig. 3 clearly indicate that our method achieves coherent retrieval performance between motion segments and textual descriptions. Additionally, we display the top-3 retrieved

motions that were unseen during training in Fig. 5, where the first two examples are from the dataset and the last example is queried using a free-form prompt.

*Table 3.* Ablation on HumanML3D.

| Methods | Text-to-motion retrieval | | | | Motion-to-text retrieval | | | |
|---|---|---|---|---|---|---|---|---|
| | R@1↑ | R@5↑ | R@10↑ | MedR↓ | R@1↑ | R@5↑ | R@10↑ | MedR↓ |
| $\lambda_S = 0$ | 10.97 | 27.79 | 39.12 | 18.00 | 11.98 | 27.67 | 38.56 | 19.50 |
| $\lambda_S = 2$ | 11.94 | 30.97 | 45.54 | 17.00 | 12.45 | 30.21 | 41.48 | 17.00 |
| $\lambda_S = 3$ | 11.23 | 28.17 | 39.94 | 18.00 | 12.36 | 28.66 | 39.44 | 18.00 |
| $\lambda_D = 0$ | 8.75 | 21.75 | 32.41 | 27.00 | 9.01 | 21.45 | 30.76 | 27.00 |
| $\lambda_D = 2$ | 11.07 | 27.52 | 39.76 | 18.00 | 11.40 | 27.23 | 38.01 | 20.00 |
| $\lambda_D = 3$ | 9.54 | 24.45 | 35.49 | 24.00 | 10.16 | 24.81 | 34.13 | 24.00 |
| $\rho_* = 0.1$ | 8.69 | 21.04 | 31.67 | 28.00 | 8.47 | 21.24 | 30.49 | 28.00 |
| $\rho_* = 0.5$ | 10.15 | 25.92 | 36.68 | 20.00 | 10.84 | 25.90 | 36.22 | 22.00 |
| $\rho_* = 0.75$ | 8.42 | 20.87 | 31.76 | 28.00 | 8.13 | 20.79 | 28.13 | 29.00 |
| Guo feature | 8.04 | 19.73 | 30.14 | 30.00 | 7.94 | 19.46 | 26.80 | 30.00 |
| Full Model | **12.45** | **33.65** | **48.22** | **10.00** | **13.59** | **31.88** | **42.68** | **15.00** |

*Table 4.* Ablation on KIT-ML.

| Methods | Text-to-motion retrieval | | | | Motion-to-text retrieval | | | |
|---|---|---|---|---|---|---|---|---|
| | R@1↑ | R@5↑ | R@10↑ | MedR↓ | R@1↑ | R@5↑ | R@10↑ | MedR↓ |
| $\lambda_S = 0$ | 14.21 | 34.87 | 51.18 | 10.00 | 13.74 | 33.73 | 45.31 | 13.00 |
| $\lambda_S = 2$ | 15.20 | 37.55 | 55.79 | 10.50 | 15.66 | 36.40 | 48.21 | 10.00 |
| $\lambda_S = 3$ | 14.98 | 35.56 | 52.23 | 9.50 | 14.08 | 34.64 | 46.51 | 12.50 |
| $\lambda_D = 0$ | 9.12 | 28.59 | 42.10 | 16.00 | 10.35 | 27.96 | 36.06 | 17.00 |
| $\lambda_D = 2$ | 13.36 | 33.28 | 48.36 | 13.00 | 13.34 | 32.83 | 44.29 | 13.00 |
| $\lambda_D = 3$ | 9.84 | 29.41 | 43.24 | 15.00 | 11.25 | 28.64 | 37.45 | 17.00 |
| $\rho_* = 0.1$ | 9.05 | 28.12 | 41.81 | 16.00 | 10.11 | 27.44 | 35.41 | 17.00 |
| $\rho_* = 0.5$ | 12.54 | 32.41 | 47.29 | 12.00 | 12.57 | 31.63 | 42.93 | 14.00 |
| $\rho_* = 0.75$ | 8.64 | 28.01 | 41.89 | 17.00 | 9.98 | 27.01 | 35.85 | 17.00 |
| Guo feature | 8.11 | 27.15 | 39.89 | 18.00 | 9.64 | 26.31 | 34.55 | 17.00 |
| Full Model | **16.01** | **38.36** | **57.87** | **7.00** | **16.76** | **37.54** | **50.17** | **8.00** |

## 4.3. Ablation Study

To validate the necessity of key components in our PST learning framework, we conduct ablation experiments focusing on four critical factors: (1) the STI distillation loss $\mathcal{L}_{SD}$ and its hyperparameter $\lambda_S$, (2) the self-distillation loss $\mathcal{L}_D$ and its hyperparameter $\lambda_D$, (3) the compression ratios in both motion and text modalities, and (4) the representation of input motion data. To comprehensively assess the contribution of each component, all ablation experiments are performed on both the HumanML3D (Guo et al., 2022) and KIT-ML (Plappert et al., 2016) datasets. As summarized in Table 3 and Table 4, each of these elements exerts a significant influence on the performance of both text-to-motion and motion-to-text retrieval tasks, clearly demonstrating their importance within our PST learning framework.

**The impact of $\mathcal{L}_{SD}$ and $\lambda_S$.** We evaluate the impact of $\mathcal{L}_{SD}$ by setting $\lambda_S$ to different values. The results show that incorporating $\mathcal{L}_{SD}$ consistently improves retrieval performance, confirming its positive contribution to motion-language retrieval. However, an increase in $\lambda_S$ leads to degraded performance, which can be attributed to the overfitting to the distillation signal.

**Text: a person swings their arms around, then walks forward and puts their arms up and down**

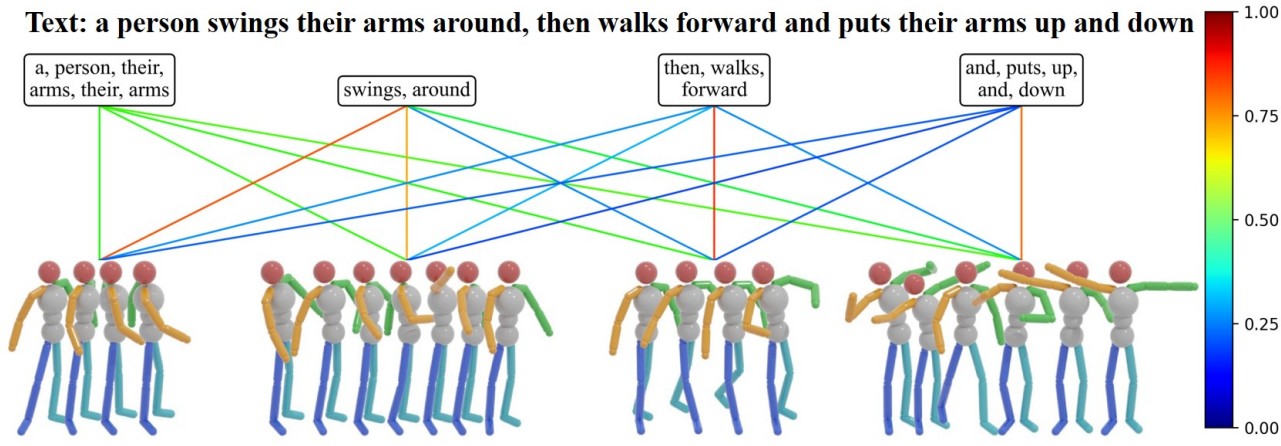

*Figure 3.* Visualization results of segment-wise alignment. We omit <EOS> for clarity and use commas to separate each individual word.

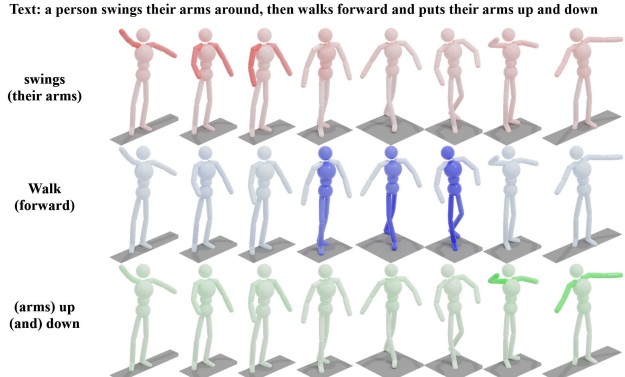

*Figure 4.* Visualization results of joint-wise alignment. Darker colors indicate higher similarity scores.

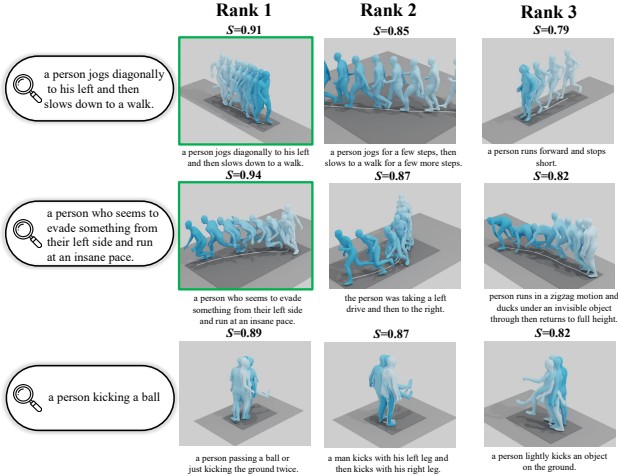

*Figure 5.* Qualitative results of text-to-motion retrieval.

**The impact of $\mathcal{L}_D$ and $\lambda_D$.** To test the critical role of $\mathcal{L}_D$, we first remove it by setting $\lambda_D = 0$, and then vary $\lambda_D$ to explore its effect on training. Consistent with the ablation results on $\mathcal{L}_{SD}$ and $\lambda_S$, incorporating $\mathcal{L}_D$ enhances both text-to-motion and motion-to-text retrieval accuracy while a high $\lambda_D$ leads to diminished performance.

**The impact of compression ratios.** We further investigate the effect of the compression ratios of motion and text tokens $\rho_*$ with default settings of $\rho_* = 0.25$ in the full model. The ablation results show that both overly low and high compression ratios degrade retrieval performance, as low ratios cause information loss while high ratios introduce redundancy and background noise.

**The impact of motion representation.** We replace our motion representation with the feature representation proposed by Guo et al. (2022), denoted as the Guo feature. Compared with the MotionPatch representation, the Guo feature results in noticeably lower recall scores. This decline can be attributed to the Guo feature's less structured representation compared with MotionPatch representation.

## 5. Conclusion

In this paper, we presented Pyramidal Shapley-Taylor (PST), a novel learning framework that goes beyond global alignment for fine-grained motion-language retrieval. Drawing inspiration from human perceptual process, PST progressively aligns motion and language through joint-wise, segment-wise, and holistic alignment stages by incorporating Shapley-Taylor Interaction (STI) and contrastive learning into training. Extensive experiments on HumanML3D and KIT-ML datasets demonstrate that PST significantly outperforms existing state-of-the-art methods on both motion-to-text and text-to-motion tasks. However, retrieval results at the joint-wise and segment-wise stages may exhibit a local bias, where the model performs particularly well on common or frequently occurring poses but struggles with complex or rare motion cases. In future work, we plan to mitigate this issue and extend PST toward open-vocabulary motion-language understanding.

## Impact Statement

This paper presents work whose goal is to advance the field of Machine Learning. There are many potential societal consequences of our work, none which we feel must be specifically highlighted here.

## Acknowledge

This work is supported in part by the National Key Research and Development Program of China (No. 2023YFC3305600), National Natural Science Foundation of China (62132016, U25B2048 and 62571393), Key Research and Development Program of Shaanxi (2024GX-YBXM-127).

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

*Figure 6.* Detailed architecture. (a) Structure of the STI Estimation Head. (b) Structure of the Token Compressor.

## A. Representation of MotionPatch

In this paper, we directly utilize the representation proposed in MotionPatch (Yu et al., 2024). Specifically, to convert a raw 3D motion sequence into a unified representation compatible with Vision Transformers, we construct motion patches by extracting spatiotemporal slices from skeleton data. Given a motion sequence $m \in \mathbb{R}^{L \times J \times 3}$, we first partition the skeleton joints into five body parts: torso, left arm, right arm, left leg and right leg, according to the human kinematic chain. Within each part, the joints are ordered by their distance to the torso to preserve structural continuity. Since different skeletons contain different numbers of joints, we apply linear interpolation along each kinematic chain to standardize each body part to $N_p$ sample points. The resulting points are normalized using z-score statistics computed over the dataset. To encode temporal information, we then take $N_p$ consecutive frames using a sliding window and stack the interpolated sample-point coordinates across these $N_p$ frames, producing an $N_p \times N_p$ spatiotemporal block. Repeating this process yields a sequence of motion patches, each analogous to an image patch, enabling motion data from diverse skeleton structures to be processed directly by ViT-based encoders.

## B. Additional Implementation Details

### B.1. Detailed Model Architecture

Our STI Estimation Head integrates local convolutional encoding with a global self-attention mechanism to enhance feature interactions across the spatiotemporal domain. Specifically, as illustrated in Fig. 6 (a), given an input tensor, this module first applies a $3 \times 3$ convolution followed by a ReLU activation to extract local representations. A self-attention layer is then applied, after which a residual connection is added between the features before and after the self-attention operation. Finally, a second $3 \times 3$ convolution is applied to refine the aggregated features.

Our Token Compressor acts as an attention-guided token compression block that performs token reduction through a combination of convolution, self-attention, and clustering. As shown in Fig. 6 (b), the Token Compressor first applies a convolutional layer with a $3 \times 1$ kernel to enrich local token interactions, followed by LayerNorm and a multi-head self-attention layer. Subsequently, each token is assigned a score via a linear projection, which is used by the KNN-DPC clustering algorithm. The resulting compressed tokens are then processed with LayerNorm and an additional self-attention layer equipped with a residual connection.

Given input embeddings $E \in \mathbb{R}^{B \times T \times D}$, where $T$ and $D$ denote the number of tokens and the feature dimension, respectively, our projection head operates along the feature dimension while preserving the token dimension. Specifically, the projection head is implemented as a two-layer feed-forward module with a GeLU, where the hidden dimension is expanded from $D$ to $2 \times D$ before being projected down to a scalar output. This design follows common practice in modern Transformer-based architectures, where a lightweight MLP head is used to map high-dimensional token embeddings into scalar scores.

### B.2. Implementation Details of Inference

#### B.2.1. PIPELINE

PST follows the standard paradigm in cross-modal retrieval and support optional fine-grained retrieval. For standard retrieval, only the holistic feature are used: each motion and text is encoded into a holistic feature after token compression, and retrieval is performed via cosine similarity between these features.

*Table 5.* Runtime and peak GPU memory consumption of different stages.

| Stage | Time (s) | Peak GPU Memory (MB) |
|---|---|---|
| Motion Encode | 0.221 | 641.25 |
| Joint-wise Alignment | 0.014 | 978.11 |
| Segment-wise Alignment | 0.011 | 342.00 |
| Holistic Alignment | 0.006 | 62.91 |
| TMR | 0.086 | 331.77 |
| MotionPatch | 0.162 | 319.65 |

*Table 6.* Ablation study on the effects of joint-wise and segment-wise alignment.

| Method | Text→Motion | | | | Motion→Text | | | |
|---|---|---|---|---|---|---|---|---|
| | R@1↑ | R@5↑ | R@10↑ | MedR↓ | R@1↑ | R@5↑ | R@10↑ | MedR↓ |
| w/o Joint-wise Alignment | 10.15 | 27.11 | 39.48 | 20.00 | 10.01 | 25.17 | 35.12 | 21.00 |
| w/o Segment-wise Alignment | 10.78 | 28.15 | 38.67 | 20.00 | 10.74 | 26.44 | 36.77 | 20.50 |
| w/o Joint-wise & Segment-wise | 8.31 | 21.78 | 32.94 | 26.00 | 9.01 | 21.85 | 31.12 | 26.50 |
| **Full Model** | **12.45** | **33.65** | **48.22** | **10.00** | **13.59** | **31.88** | **42.68** | **15.00** |

For fine-grained retrieval, joint- and segment-wise are used. Specifically, motion and text are represented by fine-grained tokens. Fine-grained retrieval is performed by computing cosine similarity between all motion–text token pairs. This fine-grained retrieval is optional and is not required for standard retrieval. Thus, PST remains efficient while enabling fine-grained retrieval.

### B.2.2. COMPUTATION COST

We further analyze the computation cost of each stage during inference compared to existing methods with same batchsize of 128. As shown in Table 5, the holistic stage used for standard retrieval is highly efficient. Although additional joint-wise and segment-wise stages are introduced, their overhead is minimal, and the overall cost is dominated by the motion encoder, which is common across methods.

### B.3. Implementation Details of Training and the STI Estimation Head

#### B.3.1. PIPELINE

For the training pipeline, our PST is trained in two stages using the same optimizer configuration, with a same batch size of 128 and 100 epochs for each stage. In the first stage, we train a teacher STI Estimation Head independently using Monte Carlo (MC) sampling to approximate the STI values based on loss function in Eq. (6). The teacher STI Estimation Head learns to match these MC–estimated interaction distributions and serves as an efficient approximation module. In the second stage, both joint-wise and segment-wise modules incorporate a student STI Estimation Head to learn which fine-grained token pairs are critical for matching. The trained teacher STI Estimation Head provides token-level supervision in the form of an STI value distribution. This interaction-based signal guides the model to focus on essential correspondences, enabling fine-grained alignment and improving retrieval performance.

#### B.3.2. ABLATION STUDY ON THREE STAGES

We validate the necessity of the pyramidal structure through stage ablation on HumanML3D datasets. As shown in Table 6, removing either the joint-wise or segment-wise alignment degrades performance, and also leads to the loss of corresponding fine-grained retrieval capability. This confirms that the pyramidal structure, when coupled with STI learning, are necessary for achieving optimal performance.

*Table 7.* Runtime and GPU memory consumption of the STI estimation head under different Monte Carlo (MC) sample numbers.

| Module / MC Samples | Time (s) | GPU Memory (MB) |
|---|---|---|
| STI Estimation Head | 0.0224 | 266.55 |
| 4 | 7.206 | 1.34 |
| 8 | 14.324 | 1.34 |
| 16 | 27.528 | 1.34 |
| **32 (default)** | **52.499** | **1.34** |
| 64 | 112.738 | 1.34 |
| 128 | 215.751 | 1.34 |

*Table 8.* Effect of the Monte Carlo (MC) sample size on cross-modal retrieval performance.

| MC Sample Size | Text→Motion | | | | Motion→Text | | | |
|---|---|---|---|---|---|---|---|---|
| | R@1↑ | R@5↑ | R@10↑ | MedR↓ | R@1↑ | R@5↑ | R@10↑ | MedR↓ |
| 4 | 2.66 | 7.12 | 12.45 | 173.00 | 4.12 | 9.01 | 14.89 | 182.50 |
| 8 | 7.35 | 19.54 | 29.84 | 27.00 | 8.71 | 10.98 | 20.36 | 30.50 |
| 16 | 10.14 | 26.68 | 39.54 | 20.00 | 10.98 | 25.66 | 36.52 | 20.00 |
| **32 (Full Model)** | **12.45** | **33.65** | **48.22** | **10.00** | **13.59** | **31.88** | **42.68** | **15.00** |
| 64 | 12.71 | 34.03 | 48.61 | 10.00 | 13.73 | 32.24 | 43.07 | 15.00 |
| 128 | 12.73 | 34.24 | 49.21 | 10.00 | 13.80 | 32.32 | 43.16 | 15.00 |

## B.4. Implementation Details of Monte Carlo Sampling

### B.4.1. IMPLEMENTATION DETAILS

Monte Carlo sampling is used to approximate the distribution of STI for each fine-grained text–motion token pair. Consequently, MC sampling is only used during the first stage of training and is not involved in inference. As describe in Sec. 3.2, we first combine text and motion tokens into a unified set. For each text–motion sample pair, we iterate over all text and motion tokens. For each selected token, we perform permutations over the unified set, compute the corresponding prefix subset, and estimate the STI value via Eq. 2, where $F(\cdot)$ is instantiated as the retrieval scoring function. The STI value for each token pair is then estimated by averaging this STI value over multiple MC sampled count, where we set to 32 in our implementation.

### B.4.2. ABLATION STUDY AND COMPUTATION COST ANALYSIS ON MC SAMPLE SIZE

We conduct an ablation study on HumanML3D datasets and a computation cost analysis on the size of MC sampling with the implementation described in Sec. 4.1. The results in Table 7 show that the runtime scales approximately linearly with the number of samples, while memory consumption remains negligible. The STI Estimation Head replaces the expensive STI with a lightweight forward pass, significantly reducing the runtime. Additionally, as shown in Table 8, increasing the sample size improves retrieval performance by providing more accurate STI supervision. However, the gain gradually saturates as the estimation becomes stable. Considering the trade-off between accuracy and computational cost, we adopt 32 MC sampling size in our implementation.

## B.5. Implementation Details of KNN-DPC

KNN-DPC is integrated into our token compressor. We take the joint-to-segment motion compression as example. For a motion sample with $N_m^{jnt} = d_j d_t$ token, where $d_j, d_t$ denote the number of joint and segment tokens, respectively, we first compress along the joint dimension via a body compressor to obtain segment-level tokens. Then we apply KNN-DPC for clustering. The number of clusters $N_C = \rho d_t$. KNN-DPC computes pairwise normalized Euclidean distances between each token to estimate local density using $k$-NN, selects cluster centers using the density-peak criterion, and assigns tokens to their nearest centers. We set $\rho = 0.25$ and $k = 5$. Finally, tokens within each cluster are merged into a representative token via a learnable linear layer. Consequently, KNN-DPC enables informative motion segments for fine-grained alignment and

*Table 9.* Motion-to-text and text-to-motion retrieval results on MotionX++.

| Method | Text→Motion | | | | | | Motion→Text | | | | | |
|---|---|---|---|---|---|---|---|---|---|---|---|---|
| | R@1↑ | R@2↑ | R@3↑ | R@5↑ | R@10↑ | MedR↓ | R@1↑ | R@2↑ | R@3↑ | R@5↑ | R@10↑ | MedR↓ |
| TEMOS(Petrovich et al., 2022) | 2.04 | 3.75 | 6.45 | 7.66 | 14.77 | 175.00 | 3.22 | 4.79 | 7.04 | 9.88 | 15.45 | 183.00 |
| T2M(Guo et al., 2022) | 1.71 | 3.20 | 4.12 | 6.07 | 11.98 | 90.50 | 3.21 | 3.54 | 6.50 | 8.45 | 13.44 | 84.00 |
| TMR(Petrovich et al., 2023) | 4.59 | 8.02 | 10.93 | 14.25 | 21.11 | 102.00 | 5.52 | 8.97 | 11.39 | 15.13 | 21.62 | 97.00 |
| MotionPatch(Yu et al., 2024) | 13.58 | 19.05 | 24.20 | 30.62 | 40.88 | 18.00 | 10.31 | 15.70 | 20.23 | 26.73 | 37.22 | 25.00 |
| Lyu et al. (2025) | 14.12 | 21.68 | 26.84 | 34.06 | 44.24 | 14.00 | 11.29 | 16.47 | 22.66 | 29.83 | 38.31 | 21.00 |
| **Ours** | **15.14** | **25.11** | **29.45** | **36.02** | **49.55** | **9.50** | **14.73** | **18.87** | **25.47** | **32.74** | **42.55** | **15.00** |

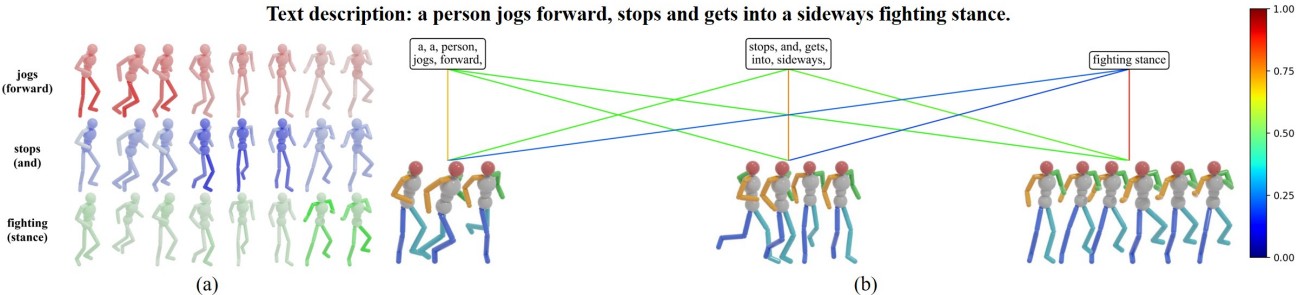

*Figure 7.* Visualization results for text description "a person jogs forward, stops and gets into a sideways fighting stance.".

provides interpretable token aggregation.

## C. Additional Experimental Results

### C.1. Additional Qualitative Results

To better demonstrate the fine-grained retrieval capability of our method, we provide additional qualitative results at both the joint-wise and segment-wise alignment stages. Concretely, we use four different text descriptions and generate four corresponding visualization results, as shown in Fig. 7, Fig. 8, Fig. 9, and Fig. 10. In each visualization, sub-figure (a) shows the joint-wise alignment results, while sub-figure (b) shows the segment-wise alignment results. Consistent with the visualizations in Sec. 4.2, sub-figure (a) in each visualization result use color intensity to represent similarity scores, where darker colors correspond to higher scores. For sub-figure (b) in each visualization, we additionally employ heatmap coloring to illustrate the similarity between each phrase and its corresponding motion segment.

### C.2. Experimental results on larger datasets

We conduct experiments on MotionX++ (Zhang et al., 2025), a significantly larger and more diverse motion-language dataset. We evaluate our method under the same experimental settings, and the results consistently demonstrate that our proposed PST framework generalizes well beyond commonly used benchmarks. The experimental results are shown in Table 9.

## D. Detailed Explanation of Equation 3

In this section, our goal is to compute the probability of different prefix subset sizes occurring under the random permutation. The Shapley-Taylor marginal contribution is computed over random permutations of all tokens in a motion-language pair. Concretely, given $N$ motion tokens and $M$ text tokens, we first concatenate them into a single unified set of size $K = N + M$. We then sample a uniformly random permutation of these $K$ tokens. For a specific motion token $m_i$ and a text token $t_j$, we denote their positions in the sampled permutation by $k$ and $p$, respectively, where $k \neq p$. We define the length $L$ of prefix subset $S_\pi$ for this pair as the number of the tokens that appear before the earliest of the motion-language pair:

$$L = \min(k, p) - 1. \tag{11}$$

**Text description: a person steps to their left, runs forward, and then back-peddles to their origin.**

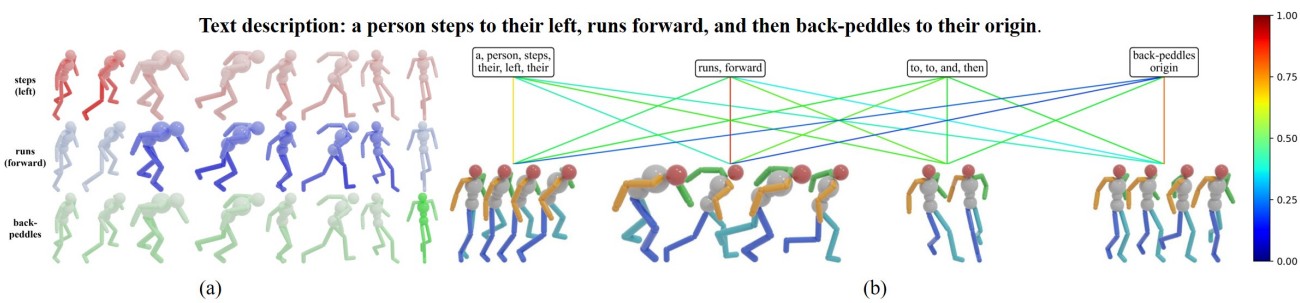

(a)                                              (b)

*Figure 8.* Visualization results for text description "a person steps to their left, runs forward, and then back-peddles to their origin.".

**Text description: person stumbles forward and bends over to pick something up.**

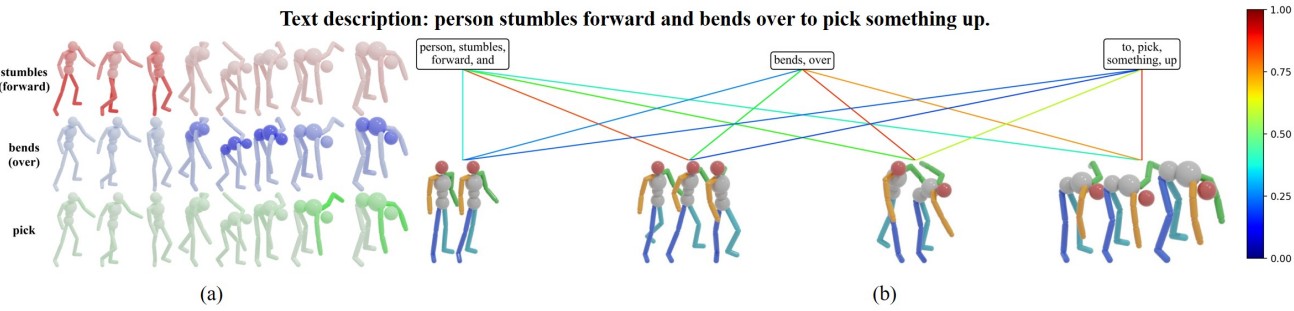

(a)                                              (b)

*Figure 9.* Visualization results for text description "person stumbles forward and bends over to pick something up.".

In other words, instead of directly reasoning about the probability of different prefix subset sizes, we reduce this problem to studying the probability distribution of the random variable $L$ induced by two positions $(k, p)$ in a random permutation. Since the permutation is uniformly random and $m_i, t_j$ are statistically independent, the joint distribution of their positions $(k, p)$ is uniform over all ordered pairs of distinct positions, indicating that there are $K(K - 1)$ ordered pairs, each with probability $1/(K(K - 1))$. We additionally define $R = \min(k, p)$, $L = R - 1$ and $r \in \{1, \dots, K - 1\}$, leading us to further reformulate the problem as probability distribution of $R$. For $R = r$ to hold, one of the tokens must appear at position $r$, and the other must appear at a position strictly after $r$. This leads to two symmetric cases:

1. $k = r$ and $p \in \{r + 1, \dots, K\}$, which yields $K - r$ valid ordered pairs.

2. $p = r$ and $k \in \{r + 1, \dots, K\}$, which also yields $K - r$ valid ordered pairs.

Therefore, the total number of ordered pairs $(k, p)$ in case of $R = r$ is $2(K - r)$, and the probability that the minimum of the two positions equals $r$ is formulated as follows:

$$\mathbb{P}(R = r) = \frac{2(K - r)}{K(K - 1)}, \quad r = 1, \dots, T - 1. \tag{12}$$

Finally, substituting $R = L + 1$ yields Equation 3

$$\begin{aligned} \mathbb{P}(L = \ell) &= \mathbb{P}(R = \ell + 1) \\ &= \frac{2(K - \ell - 1)}{K(K - 1)}. \end{aligned} \tag{13}$$

## E. Core Idea and Limitations

The key difference of PST lies in its learning paradigm. Prior methods (e.g., SGAR (Zhang et al.)) rely on explicit structural designs and supervised signals for fine-grained alignment, PST adopts a weakly supervised framework that learns

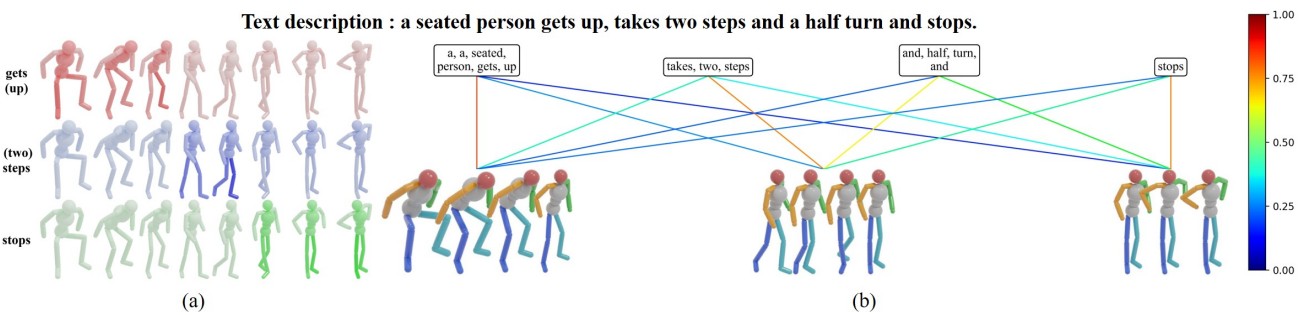

*Figure 10.* Visualization results for text description "a seated person gets up, takes two steps and a half turn and stops.".

fine-grained alignment without explicit annotations. Instead of predefined structures, PST implicitly discovers fine-grained correspondences via STI. We emphasize that PST is not intended to replace fully supervised methods, but to provide an annotation-free alternative. Its main advantage is enabling fine-grained motion-language alignment using only coarse motion-text pairs, making it suitable for weakly labeled scenarios.

A straightforward way to achieve fine-grained motion-language retrieval is to rely on explicitly annotated labels for supervised training. In contrast, our proposed PST learning framework first leverages global supervision to train the model at the holistic level, and then employs STI, which was originally developed for explainable artificial intelligence, to decompose and interpret the global retrieval process. This decomposition reveals the model's fine-grained alignment capability, which is subsequently utilized to further enhance the training of global alignment. However, this design also introduces several limitations. First, STI is used solely to attribute global similarity scores to individual tokens, and the estimator head is trained to approximate this distribution via distillation. Yet, there is no explicit supervision associating a specific word with a specific joint, meaning that local alignment is not directly constrained. As a result, the model tends to form highly consistent distributions for common motion-language patterns, leading to degraded retrieval performance on complex motions or segments with ambiguous semantic information. Moreover, in long descriptions containing repeated or highly similar motion-related phrases (e.g., multiple occurrences of word "walks"), STI-based methods exhibit a symmetry property, assigning similar attribution scores to segments with "walks" motion. Consequently, in this case, the model may struggle to differentiate which specific phrase corresponds to which motion segment, further reducing retrieval reliability in such cases. In future work, we plan to conduct a deeper investigation into these limitations and develop more principled mechanisms for fine-grained motion-language alignment.

## F. LLM-Driven Text Enhancement

During the evaluation, we found that our framework struggles to achieve precise fine-grained retrieval in more complex cases, which in turn limits the contribution of fine-grained alignment to the overall retrieval performance. A primary reason lies in the characteristics of the dataset: its textual annotations primarily describe holistic actions and lack explicit part-level descriptions of human body movements, which limits the model's ability to perform precise fine-grained retrieval. In our PST learning framework, the input motion is represented using MotionPatch (Yu et al., 2024), which provides part-level motion features, whereas the corresponding textual descriptions remain holistic, leading to a structural misalignment between the motion and text modalities. This misalignment indicates that what PST fundamentally lacks for fine-grained retrieval is an appropriate structural prior on the text modality.

In recent years, with the rapid advancement of Large Language Models (LLMs), these models have acquired rich common-sense knowledge about both human actions and their semantic descriptions. Previous work (Zhang et al.) introduces LLMs into motion-language retrieval to enrich the textual descriptions in the datasets, thereby improving the model's retrieval performance. Motivated by this observation, we attempt to enhance the textual descriptions in HumanML3D using LLMs, transforming each holistic motion description into a set of explicit, fine-grained part-level descriptions. Specifically, building upon the MotionPatch (Yu et al., 2024) representation used in our PST learning framework, we decompose the textual descriptions into five localized body parts (torso, left arm, right arm, left leg, and right leg). To generate fine-grained textual descriptions for each body part, we employ Qwen2.5-7B-Instruct as our LLM. The prompt design used for generating these part-level descriptions is illustrated in Fig. 11. Our prompt follows a progressive instruction organization. Specifically, we begin by defining the task objective, followed by explicit instructions detailing five target body parts and the constraints on

wording, granularity, and sentence structure. We then specify a strict output format and provide a concrete, high-quality example demonstrating the expected reasoning process and output style to stabilize the generation process of the LLM. In addition, all textual description of the same motion are concatenated as the input of the LLM.

HumanML3D provides multiple diverse text descriptions for each motion, offering substantially rich semantic information and allowing LLMs to generate meaningful fine-grained part-level descriptions. In contrast, the descriptions in KIT-ML are relatively simple, limiting the potential benefit of LLM-based enhancement. Therefore, in our experiments, we apply LLM-driven text enhancement only to HumanML3D. Using this enriched dataset, we retrain our model under the proposed PST learning framework and refer to this LLM-enhanced variant as PST++. During evaluation, we continue to use the original textual descriptions from HumanML3D for testing to ensure a fair and consistent comparison. Consequently, the LLM is used only for data augmentation in training stage, and the inference stage does not depend on any LLM. The quantitative results compared with previous work (Zhang et al.) on HumanML3D are demonstrated in Table 10. We directly adopt the retrieval metrics reported from SGAR (Zhang et al.). The experiment results show that enhancing the textual descriptions with LLM enables the model to learn more fine-grained motion-language correspondences, which in turn further strengthens the overall retrieval performance. The experimental results highlight the effectiveness of incorporating LLM-enhanced textual descriptions into the training process.

*Table 10.* Motion-to-text and text-to-motion retrieval results on HumanML3D with LLM-driven text enhancement.

| Protocol | Methods | Text-to-motion retrieval | | | | | | Motion-to-text retrieval | | | | | |
|---|---|---|---|---|---|---|---|---|---|---|---|---|---|
| | | R@1↑ | R@2↑ | R@3↑ | R@5↑ | R@10↑ | MedR↓ | R@1↑ | R@2↑ | R@3↑ | R@5↑ | R@10↑ | MedR↓ |
| All | SGAR (Zhang et al.) | 12.86 | - | 23.52 | 30.75 | 43.00 | 15.00 | 13.82 | - | 23.38 | 30.09 | 41.83 | 16.00 |
| | PST | 12.45 | 19.02 | 26.50 | 33.65 | 48.22 | 10.00 | 13.59 | 17.11 | 24.42 | 31.88 | 42.68 | 15.00 |
| | PST++ | **13.83** | **20.82** | **27.91** | **34.82** | **49.15** | **10.00** | **14.81** | **18.63** | **25.72** | **33.26** | **44.61** | **14.00** |
| Small batches | SGAR (Zhang et al.) | 75.66 | - | 92.11 | 95.55 | 98.06 | 1.00 | 76.35 | - | 92.38 | 95.69 | 98.02 | 1.00 |
| | PST | 76.15 | 89.31 | 94.12 | 96.35 | 98.71 | 1.00 | 75.12 | 88.79 | 93.58 | 96.64 | 98.46 | 1.00 |
| | PST++ | **78.21** | **90.15** | **95.09** | **96.71** | **98.82** | **1.00** | **76.73** | **89.11** | **94.25** | **97.18** | **98.51** | **1.00** |

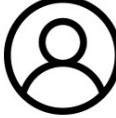

Your task is to analyze natural-language descriptions of a human motion and infer the pose and movement of each major body part. You will be given multiple text descriptions that all refer to the same motion performed by the same person. Please extract and describe the motion of each specific body part based on the combined information. Assume the performer's dominant hand is the right hand. You must describe exactly the following five body parts: [right arm, left arm, right leg, left leg, torso].
For each body part:
1. Describe its overall pose and movement throughout the complete motion.
2. Use concise, common-sense, and general wording, avoiding unnecessary details.
3. Keep each body-part description to one sentence, starting strictly with: "The [body part] ..."
Provide the final answer in the following format, with each body part in one sentence:
{
The right arm ...
The left arm ...
The right leg ...
The left leg ...
The torso ...
}
Here is an example you can refer to:
Question:
[a person runs forward and stops short.] [a person runs forward and standstill.]
Answer:
{
The right arm swings naturally back and forth while running and settles as the person stops.
The left arm swings naturally back and forth while running and settles as the person stops.
The right leg drives the run forward and then plants firmly to help halt the motion.
The left leg alternates in the running stride and braces to stabilize the sudden stop.
The torso leans slightly forward during the run and straightens as the person comes to a halt.
}
Now process the following question and provide the answer:
Q: [xxx] [xxx] ...
A:{}

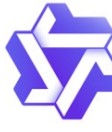

{
The right arm ...
The left arm ...
The right leg ...
The left leg ...
The torso ...
}

*Figure 11.* Text prompt for generating part-level descriptions.

