# OpenReview forum: "Beyond Global Alignment: Fine-Grained Motion-Language Retrieval via Pyramidal Shapley-Taylor Learning"
_ICML.cc/2026/Conference — ICML 2026 spotlight_

### Official Review · Reviewer_aT8y · 2026-02-23

**Soundness:** 2
**Presentation:** 2
**Significance:** 2
**Originality:** 2
**Overall Recommendation:** 4
**Confidence:** 4

**Summary:**

This paper proposed a new framework for fine-grained motion-language retrieval. Specifically, it decomposes global alignment into the three components, namely token-wise, segment-wise, and holistic alignments, and introduces Shapley-Taylor Interaction in the alignment stages to highlight motion-language correspondences. Experimental results show the effectiveness of the proposed method.

**Compliance With Llm Reviewing Policy:**

Affirmed.

**Final Justification:**

After the authors’ rebuttal, all my concerns have been successfully addressed, particularly those regarding the previously unclear technical details and missing experimental results. Hence, I am willing to increase my score accordingly. I encourage the authors to include these improvements in the final version and further polish the manuscript for clearer presentation.

**Key Questions For Authors:**

1. What is the pipeline during the inference stage? Are token-wise, segment-wise, and holistic features all used, or is only the holistic feature used via similarity matching for inference?
2. Is the training of STI Estimation Head $\mathcal{H}$ via Monte Carlo sampling independent of the alignment training stage? If so, the hyperparameter choices for Monte Carlo sampling and more implementation details of $\mathcal{H}$ are not sufficiently justified, despite being central to Eq. (6). Additionally, the computational cost should be reported.
3. Does KNN-DPC introduce additional training time? Moreover, the effectiveness and necessity of this module should be evaluated through ablation studies. In addition, the joint-wise, segment-wise, and holistic alignments are not individually evaluated in the experiments.
4. Why do the authors use the joint-wise alignment stage as teacher signals instead of the segment-wise stage? This choice lacks both empirical evidence and theoretical justification.

**Limitations:**

The authors are encouraged to discuss the limitations of the proposed method.

**Strengths And Weaknesses:**

**Strengths**:
1. Introducing Shapley-Taylor Interaction (STI) to capture token-level motion-language correspondence is an interesting idea. Moreover, the three-stage fine-grained alignment provides new insights for the community.
2. The visualization figures are intuitive and clear, and the experimental results show the effectiveness of the proposed method.

**Weaknesses**:
1. The paper is not well-written, as many technical details are insufficiently justified or only briefly described, such as the Monte Carlo sampling of STI and KNN-DPC.
2. The experimental section is not comprehensive, lacking ablation studies on several critical modules as well as an evaluation of the necessity of the three alignment stages.
3. The proposed framework is relatively complex, involving multiple additional loss function designs, while the performance improvement is marginal, which raises concerns about its practical value.

---

> ### Author Rebuttal · Authors · 2026-03-31
>
> # Response To Reviewer aT8y:
> ## Thanks for your valuable suggestions. We will incorporate relevant explanation into the final version.
>
> # Q1: Details of inference
> ## Pipeline
>
> PST follows the standard paradigm in cross-modal retrieval and support optional fine-grained retrieval. For standard retrieval, only the holistic feature are used: each motion and text is encoded into a holistic feature after token compression, and retrieval is performed via cosine similarity between these features.
>
> For fine-grained retrieval, joint- and segment-wise are used. Specifically, motion and text are represented by fine-grained tokens. Fine-grained retrieval is performed by computing cosine similarity between all motion–text token pairs. This fine-grained retrieval is optional and is not required for standard retrieval. Thus, PST remains efficient while enabling fine-grained retrieval.
>
> ## Computation cost
> We kindly refer the reviewer to our response to Q2 of Reviewer rAmJ for details.
>
>
> ## Model complexity and practical benefits of PST
> We agree that PST introduces additional modules, but this complexity arises from fine-grained modeling, which cannot be captured by global-only methods. PST decomposes alignment into three stages, each capturing complementary information at different scales. STI models motion–text interactions, enabling weakly supervised fine-grained alignment without explicit annotations. Thus, PST learns fine-grained correspondences beyond conventional methods. Importantly, the contribution of PST lies in enabling interpretable and weakly supervised fine-grained modeling. Although gains on standard metrics are moderate, PST provides fine-grained retrieval capability and interpretability.
>
>
> # Q2: Details of training and STI Estimation Head
> ## Pipeline
> We kindly refer the reviewer to our response to Q3 of Reviewer rAmJ for details.
>
> ## Ablation study on three stages
> We kindly refer the reviewer to our response to Q1 of Reviewer iBut for details.
>
> # Q3: Details of Monte Carlo (MC) Sampling
> ## Implementation Details
> We kindly refer the reviewer to our response to Q1 of Reviewer rAmJ for details.
>
> ## Ablation study and computation cost analysis on MC Sample size
> We kindly refer the reviewer to our response to Q2 of Reviewer 6aDZ for details.
>
>
> # Q4: Details of KNN-DPC
> ## Implementation details
>
> KNN-DPC is integrated into our token compressor. We take the joint-to-segment motion compression as example. For a motion sample with $N_m^{jnt}=d_j × d_t$ token, where $d_j, d_t$ denote the number of joint and segment tokens, respectively, we first compress along the joint dimension via a body compressor to obtain segment-level tokens. Then we apply KNN-DPC for clustering. The number of clusters $N_C$ is $N_C=\rho × d_t$. KNN-DPC computes pairwise normalized Euclidean distances between each token to estimate local density using $k$-NN, selects cluster centers using the density-peak criterion, and assigns tokens to their nearest centers. We set $\rho=0.25$ and $k=5$. Finally, tokens within each cluster are merged into a representative token via a learnable linear layer. Consequently, KNN-DPC enables informative motion segments for fine-grained alignment and provides interpretable token aggregation.
>
> ## Computation cost
> We acknowledge that KNN-DPC introduces additional cost. However, as shown below, the cost is negligible overall.
>
> |Module|Time(s)|GPU Memory(MB)|
> |-|-|-|
> |KNN-DPC|0.0006|0.69|
> |Token Compressor|0.0021|4.53|
> |PST|0.302|1531.55|
>
> ## Ablation study
> By density-based clustering, KNN-DPC provides an interpretable aggregation and preserve structured motion features, while operations such as conv/pooling may mix features implicitly and lead to information loss. We conduct an ablation study on HumanML3D by replacing KNN-DPC with convolution and average pooling. As shown below, replacing KNN-DPC leads to performance degradation.
>
> |Module|(Text→Motion)R@1↑|R@5↑|R@10↑|MedR↓|(Motion→Text)R@1↑|R@5↑|R@10↑|MedR↓|
> |-|-|-|-|-|-|-|-|-|
> |avg pool|8.42|21.41|31.83|26.00|9.06|22.37|31.67|26.50|
> |conv|9.15|22.56|33.14|25.50|9.79|23.22|33.78|26.00|
> |**KNN-DPC**|**12.45**|**33.65**|**48.22**|**10.00**|**13.59**|**31.88**|**42.68**|**15.00**|
>
> # Q5: Reason of joint-wise as teacher signals.
>
> The self-distillation loss aligns distributions across levels, preventing higher-level  drifting. We use the joint-wise as the teacher for two reasons. (1) it follows the bottom-up flow of PST, where segment features are derived from joints. (2) joint-wise retains finer details, while segment-wise involves aggregation that may cause information loss. Therefore, joint-wise supervision better preserves detailed alignment. Ablation on HumanML3D shows segment-wise teacher degrades performance.
>
> |Module|(Text→Motion)R@1↑|R@5↑|R@10↑|MedR↓|(Motion→Text)R@1↑|R@5↑|R@10↑|MedR↓|
> |-|-|-|-|-|-|-|-|-|
> |segment as teacher|8.72|21.11|35.89|25.50|8.23|23.41|32.94|26.00|
> |**Full**|**12.45**|**33.65**|**48.22**|**10.00**|**13.59**|**31.88**|**42.68**|**15.00**|

---

> > ### Author Rebuttal · Reviewer_aT8y · 2026-04-01
> >
> > After the authors’ rebuttal, all my concerns have been successfully addressed, particularly those regarding the previously unclear technical details and missing experimental results. I encourage the authors to include these improvements in the final version and further polish the manuscript for clearer presentation.

---

### Official Review · Reviewer_rAmJ · 2026-03-06

**Soundness:** 3
**Presentation:** 3
**Significance:** 3
**Originality:** 3
**Overall Recommendation:** 4
**Confidence:** 4

**Summary:**

This paper proposes a novel Pyramidal Shapley-Taylor (PST) learning framework for fine-grained motion–language retrieval, addressing the limitations of traditional global alignment methods. Inspired by the process of human motion perception, PST process retrieval in joint-wise alignment, segment-wise alignment and holistic alignment stages. Concretely, the authors introduce a lightweight STI Estimation Head trained via Monte-Carlo distillation to approximate STI signals during training. Between each adjacent alignment stage, a learnable compressor implemented with KNN-DPC clustering is utilized to aggregate coherent tokens. Extensive experiments on HumanML3D and KIT-ML datasets demonstrate state-of-the-art performance on both text-to-motion and motion-to-text retrieval, and the supplementary material further strengthens the work with detailed architectural descriptions, mathematical derivations, additional qualitative analyses. An extended variant, PST++, which incorporates LLM-driven text enhancement, provides additional insights into the effectiveness and extensibility of this framework.

**Compliance With Llm Reviewing Policy:**

Affirmed.

**Final Justification:**

The rebuttal addressed my concerns.

**Key Questions For Authors:**

1.How much computational saving is achieved by the Monte Carlo STI approximation? What is the computational overhead introduced by PST in each alignment stage?

2.How is Monte Carlo sampling concretely applied when training the STI estimation head, and what are the practical settings (e.g., number of permutations and samples)?

3.How the number of Monte Carlo samples affects the performance of the PST learning framework.

4.What does “optimized separately” mean for the STI Estimation Head and the retrieval model? Does this refer to sequential pretraining, alternating optimization, or joint training with separate losses within each iteration?

5.How does the prefix-length distribution in Sec. 3.2 lead to STI peaks for genuinely aligned motion–language pairs, and how do these peaks influence motion–language alignment? Can the authors provide a more intuitive explanation of this mechanism?

**Limitations:**

Yes

**Strengths And Weaknesses:**

Strength:

1.The paper introduces a pyramidal learning framework that progressively models motion–language correspondences from joint-wise to segment-wise and holistic levels, closely mirroring the hierarchical nature of human motion perception and enabling coarse-to-fine semantic learning beyond conventional global alignment, which is technically sound.

2.The integration of Shapley-Taylor Interaction (STI) enables the model to capture hierarchical relationships between motion and language that are overlooked by global-centric methods, thereby improving the accuracy of motion–language retrieval and achieving state-of-the-art performance, which demonstrates the significance of the proposed approach.

3.The qualitative visualization results of joint-wise alignment and segment-wise alignment provide an intuitive presentation of this work, which helps readers understand the fine-grained cross-modal correspondences and supports the effectiveness of this pyramidal learning framework design.

4.The paper conducts comprehensive ablation studies on the key components of the PST learning framework, which provides solid support for the overall architectural design.

5.The PST++ variant in supplementary material demonstrates that enhancing textual structural granularity via LLM further improves fine-grained retrieval, highlighting the practical applicability of the proposed framework.

Weaknesses:

1.Computational and efficiency aspects of PST are not analyzed. Although STI is approximated via Monte Carlo sampling and an estimator to reduce the cost of random permutations, the paper does not quantify the computational savings or report the resulting training time and overhead of the overall framework.

2.Some implementation details of the STI-based training are under-specified. The paper does not clearly describe how Monte Carlo sampling is applied to train the STI estimation head or how many permutations are sampled, which may hinder reproducibility.

3.Although PST technically sound, the core ideas are not sufficiently emphasized, which may lead to misunderstanding. The key distinction of PST learning framework lies in obtaining fine-grained alignment signals through unsupervised STI-based distillation without explicit local annotations; however, this difference from prior structured alignment methods, such as SGAR, is not sufficiently emphasized throughout the paper, which may lead to misunderstanding from readers.

4.Minor inconsistency in training description. The paper states that the STI Estimation Head and retrieval model are “optimized separately” (Sec. 4.1), but it is unclear whether this means sequential pretraining, alternating optimization, or separate losses in the same iteration.

5.The explanation of the prefix-length weighting in Sec. 3.2 is somewhat abstract and hard to follow. The connection between shorter prefixes and STI peaks for true correspondences is not clearly illustrated, which may hinder intuitive understanding.

---

> ### Author Rebuttal · Authors · 2026-03-30
>
> # Response To Reviewer rAmJ:
> ## Thanks for your valuable suggestions. We will incorporate relevant explanation into the final version for better understanding
>
> # Q1: Details of Monte Carlo (MC) Sampling
> ## Implementation details
>
> Monte Carlo sampling is used to approximate the distribution of STI for each fine-grained text–motion token pair. Consequently, MC sampling is only used during the first stage of training and is not involved in inference.
>
> As describe in Sec. 3.2, we first combine text and motion tokens into a unified set. For each text–motion sample pair, we iterate over all text and motion tokens. For each selected token, we perform permutations over the unified set, compute the corresponding prefix subset, and estimate the STI value via Eq. (2), where $F(⋅)$ is instantiated as the retrieval scoring function. The STI value for each token pair is then estimated by averaging this STI value over multiple MC sampled count, where we set to 32 in our implementation.
>
> ## Ablation study and computation cost analysis on MC Sample size
> We kindly refer the reviewer to our response to Q2 of Reviewer 6aDZ for more details.
>
>
> # Q2: The detailed computation cost in PST.
> We further analyze the computation cost of each stage during inference compared to existing methods with same batchsize of 128. As shown below, the holistic stage used for standard retrieval is highly efficient. Although additional joint-wise and segment-wise stages are introduced, their overhead is minimal, and the overall cost is dominated by the motion encoder, which is common across methods.
>
> |Stage|Time(s)|Peak GPU Memory(MB)|
> |-|-|-|
> |Motion Encode|0.221|641.25|
> |Joint-wise alignment|0.014|978.11|
> |Segment-wise alignment|0.011|342.00|
> |Holistic alignment|0.006|62.91|
> |TMR|0.086|331.77|
> |MotionPatch|0.162|319.65|
>
>
> # Q3: Detail of training pipeline and STI Estimation Head.
> For the detailed architecture of STI Estimation Head, please refer to SupMat.
>
> We sincerely apologize for any confusion it may have caused. In our paper, "optimized separately" refers to a two-stage training procedure. For the training pipeline, our PST is trained in two stages using the same optimizer configuration, with a same batch size of 128 and 100 epochs for each stage. In the first stage, we train a teacher STI Estimation Head independently using Monte Carlo (MC) sampling to approximate the STI values based on loss function in Eq. (6). The teacher STI Estimation Head learns to match these MC–estimated interaction distributions and serves as an efficient approximation module. In the second stage, both joint-wise and segment-wise modules incorporate a student STI Estimation Head to learn which fine-grained token pairs are critical for matching. The trained teacher STI Estimation Head provides token-level supervision in the form of an STI value distribution. This interaction-based signal guides the model to focus on essential correspondences, enabling fine-grained alignment and improving retrieval performance.
>
>
> # Q4: Core Idea and Distinction from Prior Work
> We appreciate the reviewer's suggestion. The key difference of PST lies in its learning paradigm. Prior methods (e.g., SGAR) rely on explicit structural designs and supervised signals for fine-grained alignment, PST adopts a weakly supervised framework that learns fine-grained alignment without explicit annotations. Instead of predefined structures, PST implicitly discovers fine-grained correspondences via STI. We emphasize that PST is not intended to replace fully supervised methods, but to provide an annotation-free alternative. Its main advantage is enabling fine-grained motion-language alignment using only coarse motion-text pairs, making it suitable for weakly labeled scenarios. We will clarify this in the final version.
>
> # Q5: Intuition Behind Prefix-Length Weighting and STI Peaks
> Eq. (3) defines the probability distribution of prefix lengths. When the prefix set $S_{\pi}$ is small, adding a matched pair yields a large marginal gain, resulting in high STI values. In contrast, when $S_{\pi}$ is large, redundant tokens reduce the additional contribution of new pairs, leading to lower STI values. This mechanism directly benefits motion–language alignment. STI peaks enable more discriminative retrieval by highlighting the most informative token pairs under limited context, while suppressing spurious or redundant correlations. Through distillation, they further provide fine-grained supervision that encourages the model to focus on truly aligned token pairs, leading to more accurate similarity estimation and improved retrieval performance.

---

> > ### Author Rebuttal · Reviewer_rAmJ · 2026-04-02
> >
> > My concerns have been adequately addressed

---

### Official Review · Reviewer_uZif · 2026-03-12

**Soundness:** 3
**Presentation:** 2
**Significance:** 2
**Originality:** 2
**Overall Recommendation:** 5
**Confidence:** 4

**Summary:**

This paper takes inspiration from the pyramidal process theorized in human motion perception and proposes a novel Pyramidal Shapley-Taylor (PST) learning framework tailored for fine-grained motion-language retrieval. The PST framework decomposes human motion data into both temporal segments and spatial body joints, learning cross-modal correspondences through a progressive alignment scheme at joint and segment levels. By employing a hierarchical, pyramidal alignment, the method aims to capture both local semantic details and higher-level structural relationships for improved motion-language retrieval.

**Compliance With Llm Reviewing Policy:**

Affirmed.

**Final Justification:**

Thanks for the effort on the rebuttal. The response has fully addressed my concerns, especially by providing the source of the motivation and more experimental results. Hence, I am inclined to recommend acceptance of this work.

**Key Questions For Authors:**

As mentioned in the weaknesses.

**Limitations:**

As mentioned in the weaknesses.

**Strengths And Weaknesses:**

## Strengths

1. The proposed Pyramidal Shapley-Taylor (PST) learning framework is conceptually novel, and the idea of multi-level progressive cross-modal alignment may offer new insights for motion-language retrieval.
2. The paper is well-organized and clearly written, making the proposed ideas and methodology easy to follow.

## Weaknesses

1. The claim that human motion perception unfolds in three sequential cognitive stages is speculative and not supported by citations or authoritative sources. Unless the authors can provide strong empirical or scientific evidence, this foundational motivation remains unconvincing.
2. The core technical novelty is limited. While PST introduces a hierarchical learning structure, it essentially operates as a standard pyramidal or multi-stage training approach, which is commonly used. The absence of ablation studies comparing PST to conventional pyramid structures makes it unclear how much the proposed method itself contributes.
3. The comparative experiments primarily reference older works. A stronger experimental section would benchmark PST against more recent and competitive motion-language retrieval methods.
4. The manuscript would benefit from a clearer articulation of the unique value and trade-offs of retrieval-based motion approaches compared to generative ones. Specifically, issues such as diversity and fine-grained textual control should be discussed.

---

> ### Author Rebuttal · Authors · 2026-03-30
>
> # Response To Reviewer uZif:
> ## Thanks for your valuable suggestions. We will incorporate relevant explanation into the final version for better understanding
>
> # Q1: The claim of human motion perception.
> The inspiration of human motion perception was draw from prior findings in biological motion perception. Giese et al. [1] show that biological motion recognition relies on two types, structural information, such as joint and posture, and dynamic information, such as joint trajectories. These two types are processed and integrated at higher levels neuro to enable motion recognition. Similarly, Troje et al. [2] emphasizes that biological motion perception involves multiple levels of processing, including local motion cues, structure of motion and high-level interpretation. This suggests that motion understanding requires both local and global signals. Consequently, PST adopts a pyramidal design as a computational abstraction of multi-level motion processing.
>
> In addition, this multi-level motivation is also aligned with recent trends in motion realm. Li et al. [3] combines global sequence-level text with local frame-level text and emphasizes hierarchical modeling, while Fang et al. [4] highlights the importance of fine-grained interaction modeling for motion understanding. These works support the trend toward fine-grained modeling in motion-language tasks.
>
> Overall, PST is motivated not only by biological insights, but also by trends in motion realm that increasingly emphasize fine-grained modeling.
>
> [1] Giese et al.. Neural mechanisms for the recognition of biological movements. (Nature Reviews Neuroscience)
>
> [2] Troje et al.. Biological motion perception. (The senses: A comprehensive reference)
>
> [3] Li et al.. Unimotion: Unifying 3d human motion synthesis and understanding. (3DV2025)
>
> [4] Fang et al.. Humocon: Concept discovery for human motion understanding. (CVPR2025)
>
> # Q2: Concern of the pyramidal novelty.
> ## Response to the concern on pyramidal novelty.
> We agree that simply using a naïve pyramidal structure is not novel. However, the novelty of PST lies not in the structure itself, but in its coupling with STI learning for fine-grained motion-language retrieval. Specifically, PST decomposes alignment into three stages, each capturing complementary information at different granularities. This pyramidal structure is not a standalone design, it provides the essential representation that STI relies on to model fine-grained motion–text interactions. Additionally, STI is not an auxiliary component, but a key mechanism that leverages this pyramidal structure to models motion–text interactions and enable weakly supervised fine-grained alignment without explicit annotations. Through pyramidal structure and STI, PST learns fine-grained correspondences that are not accessible in conventional frameworks. Importantly, the contribution of PST lies in enabling interpretable and weakly supervised fine-grained modeling.
>
>
> ## Ablation study on three stages of pyramidal structure.
> We kindly refer the reviewer to our response to Q1 of Reviewer iBut for more details.
>
> # Q3: Experiment results compared to more recent methods.
>
> We include additional comparisons with several recent and competitive approaches on HumanML3D datasets, including KinMo [1], CAR [2] and SGAR, where the comparison with SGAR is shown in SupMat. As shown below, PST consistently outperforms these methods.
>
> |Method|(Text→Motion)R@1↑|R@2↑|R@3↑|R@5↑|R@10↑|MedR↓|(Motion→Text)R@1↑|R@2↑|R@3↑|R@5↑|R@10↑|MedR↓|
> |-|-|-|-|-|-|-|-|-|-|-|-|-|
> |CAR|8.03|14.51|18.84|26.73|38.98|17.00|11.72|15.19|21.65|28.15|39.23|17.50|
> |KinMo|9.05|15.23|20.47|28.62|41.60|16.00|9.01|15.92|21.42|29.50|41.43|16.00|
> |**PST (Ours)**|**12.45**|**19.02**|**26.50**|**33.65**|**48.22**|**10.00**|**13.59**|**17.11**|**24.42**|**31.88**|**42.68**|**15.00**|
>
> [1] Zhang et al.. Kinmo: Kinematic-aware human motion understanding and generation. (ICCV2025)
>
> [2] Fujiwara et al.. Chronologically accurate retrieval for temporal grounding of motion-language models. (ECCV2024)
>
> # Q4: The trade-offs of retrieval approaches compared to generative ones.
> Retrieval-based and generative methods serve complementary roles in motion-language modeling. Retrieval-based approaches select motions from real database, ensuring strong realism and physical plausibility, while generative methods provide greater diversity and flexibility. We acknowledge that generative models offer stronger diversity and controllability. However, achieving precise and stable local control remains challenging in practice. In contrast, retrieval-based methods can provide more reliable and interpretable correspondence at the local database. Our work focuses on strengthening the retrieval paradigm by addressing limitation in fine-grained alignment, instead of replacing generative approaches.

---

> > ### Author Rebuttal · Reviewer_uZif · 2026-04-01
> >
> > Thanks for the effort on the rebuttal. The response has fully addressed my concerns, especially by providing the source of the motivation and more experimental results. Hence, I am inclined to recommend acceptance of this work.

---

### Official Review · Reviewer_iBut · 2026-03-13

**Soundness:** 3
**Presentation:** 3
**Significance:** 3
**Originality:** 3
**Overall Recommendation:** 5
**Confidence:** 4

**Summary:**

This paper studies motion-language retrieval and argues that existing methods mainly focus on global alignment between entire motion sequences and text descriptions. The authors propose a Pyramidal Shapley-Taylor framework that introduces hierarchical alignment across joint-wise, segment-wise, and holistic levels. Experiments on HumanML3D and KIT-ML show modest improvements over prior retrieval methods.

**Compliance With Llm Reviewing Policy:**

Affirmed.

**Key Questions For Authors:**

See Weaknesses.

**Limitations:**

yes

**Strengths And Weaknesses:**

Strengths:
- The paper tackles an important problem in motion–language understanding. The motivation is clear and well explained. Modeling fine-grained correspondences between motion and text is a natural and meaningful direction.

- The pyramidal alignment idea is simple but effective. Modeling interactions from joints to segments and then to the full sequence is intuitive and well justified. This design makes the framework easy to understand.

- The empirical results are good. The method consistently outperforms prior approaches on both HumanML3D and KIT-ML. The improvements are stable across metrics and evaluation protocols.

Weaknesses:
- The necessity of the pyramidal alignment structure is not convincingly demonstrated.

- The experiments are mainly on two commonly used datasets. Additional evaluations on larger or more diverse datasets could further strengthen the empirical evidence.

---

> ### Author Rebuttal · Authors · 2026-03-30
>
> # Response To Reviewer iBut:
> ## Thanks for your valuable suggestions. We will incorporate relevant explanation into the final version for better understanding
>
> # Q1: The necessity of the pyramidal structure.
> ## Claim of the necessity.
> The inspiration of human motion perception was draw from prior findings in neuroscience and biological motion perception. Specifically, Giese et al. [1] show that biological motion recognition relies on two distinct types, structural information, such as joint configuration and posture, and dynamic information, such as joint trajectories. These two types of information are processed and integrated at higher levels neuro to enable motion recognition. Similarly, Troje et al. [2] emphasizes that biological motion perception is not a global process, but involves multiple levels of processing, including local motion cues, structure from motion and higher-level motion interpretation. This multi-level perspective suggests that effective motion understanding requires modeling both fine-grained local signals and global representations. Inspired by this research, our PST framework adopts a pyramidal design as a computational abstraction of multi-level motion processing, where different stages capture complementary information at different levels of granularity.
>
> In addition, this multi-level motivation is also aligned with recent trends in motion realm. Li et al. [3] explicitly combines global sequence-level text with local frame-level text and emphasizes hierarchical, time-aware motion understanding, while Fang et al. [4] highlights the importance of fine-grained interaction modeling for human motion understanding. These works do not study fine-grained retrieval directly, but they support the trend that effective motion-language modeling increasingly benefits from fine-grained modeling.
>
> Overall, we emphasize that the design of PST is not only motivated by biological studies on multi-level human motion perception, but is also aligned with recent trends in motion realm, which increasingly highlight the importance of fine-grained modeling.
>
> [1] Giese et al.. Neural mechanisms for the recognition of biological movements. (Nature Reviews Neuroscience)
>
> [2] Troje et al.. Biological motion perception. (The senses: A comprehensive reference)
>
> [3] Li et al.. Unimotion: Unifying 3d human motion synthesis and understanding. (3DV2025)
>
> [4] Fang et al.. Humocon: Concept discovery for human motion understanding. (CVPR2025)
>
> ## Ablation study on three stages of pyramidal structure.
> We validate the necessity of the pyramidal structure through stage ablation on HumanML3D datasets. As shown below, removing either the joint-wise or segment-wise alignment degrades performance, and also leads to the loss of corresponding fine-grained retrieval capability. This confirms that the pyramidal structure, when coupled with STI learning, are necessary for achieving optimal performance.
>
> |Method|(Text→Motion)R@1↑|R@5↑|R@10↑|MedR↓|(Motion→Text)R@1↑|R@5↑|R@10↑|MedR↓|
> |-|-|-|-|-|-|-|-|-|
> |w/o Joint-wise Alignment|10.15|27.11|39.48|20.00|10.01|25.17|35.12|21.00|
> |w/o Segment-wise Alignment|10.78|28.15|38.67|20.00|10.74|26.44|36.77|20.50|
> |w/o Joint-wise & Segment-wise|8.31|21.78|32.94|26.00|9.01|21.85|31.12|26.50|
> |**Full Model**|**12.45**|**33.65**|**48.22**|**10.00**|**13.59**|**31.88**|**42.68**|**15.00**|
>
> # Q2: Experimental results on larger datasets.
>
> We conduct experiments on MotionX++, a significantly larger and more diverse motion-language dataset. We evaluate our method under the same experimental settings, and the results consistently demonstrate that our proposed PST framework generalizes well beyond commonly used benchmarks. We will include these additional results in the final version to further support the scalability of our method.
>
> |Method|(Text→Motion)R@1↑|R@2↑|R@3↑|R@5↑|R@10↑|MedR↓|(Motion→Text)R@1↑|R@2↑|R@3↑|R@5↑|R@10↑|MedR↓|
> |-|-|-|-|-|-|-|-|-|-|-|-|-|
> |TEMOS|2.04|3.75|6.45|7.66|14.77|175|3.22|4.79|7.04|9.88|15.45|183.00|
> |T2M|1.71|3.20|4.12|6.07|11.98|90.50|3.21|3.54|6.50|8.45|13.44|84.00|
> |TMR|4.59|8.02|10.93|14.25|21.11|102.0|5.52|8.97|11.39|15.13|21.62|97.00|
> |MotionPatch|13.58|19.05|24.20|30.62|40.88|18.00|10.31|15.70|20.23|26.73|37.22|25.00|
> |Lyu et al.|14.12|21.68|26.84|34.06|44.24|14.00|11.29|16.47|22.66|29.83|38.31|21.00|
> |**Ours**|**15.14**|**25.11**|**29.45**|**36.02**|**49.55**|**9.50**|**14.73**|**18.87**|**25.47**|**32.74**|**42.55**|**15.00**|

---

> > ### Author Rebuttal · Reviewer_iBut · 2026-04-03
> >
> > The authors’ rebuttal satisfactorily addresses my concerns, so I maintain my original accept score.

---

### Official Review · Reviewer_6aDZ · 2026-03-19

**Soundness:** 4
**Presentation:** 3
**Significance:** 4
**Originality:** 4
**Overall Recommendation:** 4
**Confidence:** 5

**Summary:**

The author introduces a novel Pyramidal Shapley-Taylor (PST) framework for fine-grained motion–language retrieval. By conducting hierarchical alignment at the joint, segment, and holistic levels, it addresses the shortcomings of conventional global alignment approaches. The method incorporates a lightweight STI estimation head trained through Monte Carlo distillation, along with a learnable compressor based on KNN-DPC clustering between each alignment stages. In general, leveraging Shapley-Taylor to model and interpret cross-modal interactions is conceptually compelling and well-motivated.

**Compliance With Llm Reviewing Policy:**

Affirmed.

**Key Questions For Authors:**

1.Could you clarify the roles of the retrieval model and the STI model in PST, and how they interact within the overall training and inference pipeline?
2.Could you provide a brief discussion on how PST can be adapted to other motion representations instead of MotionPatch?
3. Could you clarify the computational cost and efficiency of the method, as well as provide more details on the STI approximation (e.g., number of sampled permutations and Monte Carlo sampling), to improve reproducibility?

**Limitations:**

YES

**Strengths And Weaknesses:**

Strengths:

1.The pyramidal design effectively addresses the lack of fine-grained alignment in prior work. The pipeline figure is highly informative and intuitive.
2.By introducing STI, the model captures hierarchical cross-modal interactions that global approaches miss, leading to improved retrieval performance.
3.The joint- and segment-level visualizations clearly illustrate fine-grained alignments, making the method more intuitive and interpretable.
4.The PST++ extension demonstrates that enhancing the structural complexity of text with LLMs can further improve fine-grained retrieval performance, while maintaining efficient inference since the LLM is only involved during training.


Weaknesses:
1.The paper does not analyze computational cost or efficiency. Although STI is approximated via Monte Carlo sampling and an estimator, there is no discussion of training overhead, runtime, or actual savings.
2.The paper lacks sufficient clarity regarding the practical implementation of STI computation and its approximation. In particular, aspects such as the number of sampled permutations, and the use of Monte Carlo sampling are not fully specified, which may affect reproducibility.
3.The distinction between the retrieval model and the STI model mentioned in experimental section is unclear, and their training procedure is somewhat ambiguous, which may lead to confusion for readers.

---

> ### Author Rebuttal · Authors · 2026-03-30
>
> # Response To Reviewer 6aDZ:
> ## Thanks for your valuable suggestions. We will incorporate relevant explanation into the final version for better understanding
>
> # Q1: The computation cost of PST
> We analyze the computation cost of each stage during inference compared to existing methods with same batchsize of 128. As shown below, the holistic stage used for standard retrieval is highly efficient. Although additional joint-wise and segment-wise stages are introduced, their overhead is minimal, and the overall cost is dominated by the motion encoder, which is common across methods.
>
> |Stage|Time(s)|Peak GPU Memory(MB)|
> |-|-|-|
> |Motion Encode|0.221|641.25|
> |Joint-wise alignment|0.014|978.11|
> |Segment-wise alignment|0.011|342.00|
> |Holistic alignment|0.006|62.91|
> |TMR|0.086|331.77|
> |MotionPatch|0.162|319.65|
>
>
> # Q2: Ablation study and computation cost analysis on MC Sampling size
>
> We conduct an ablation study on HumanML3D datasets and a computation cost analysis on the size of MC sampling with the implementation described in Sec. 4.1. The results below show that the runtime scales approximately linearly with the number of samples, while memory consumption remains negligible. The STI Estimation Head replaces the expensive STI with a lightweight forward pass, significantly reducing the runtime. Additionally, increasing the sample size improves retrieval performance by providing more accurate STI supervision. However, the gain gradually saturates as the estimation becomes stable. Considering the trade-off between accuracy and computational cost, we adopt 32 MC sampling size in our implementation. Detailed results will be included in the final version.
>
> |Module/MC Samples|Time(s)|GPU Memory(MB)|
> |-|-|-|
> |STI Estimation Head|0.0224|266.55|
> |4|7.206|1.34|
> |8|14.324|1.34|
> |16|27.528|1.34|
> |**32(default)**|52.499|1.34|
> |64|112.738|1.34|
> |128|215.751|1.34|
>
> |MC Sample Size|(Text→Motion)R@1↑|R@5↑|R@10↑|MedR↓|(Motion→Text)R@1↑|R@5↑|R@10↑|MedR↓|
> |-|-|-|-|-|-|-|-|-|
> |4|2.66|7.12|12.45|173.00|4.12|9.01|14.89|182.50|
> |8|7.35|19.54|29.84|27.00|8.71|10.98|20.36|30.50|
> |16|10.14|26.68|39.54|20.00|10.98|25.66|36.52|20.00|
> |**32(Full Model)**|**12.45**|**33.65**|**48.22**|**10.00**|**13.59**|**31.88**|**42.68**|**15.00**|
> |64|12.71|34.03|48.61|10.00|13.73|32.24|43.07|15.00|
> |128|12.73|34.24|49.21|10.00|13.80|32.32|43.16|15.00|
>
>
> # Q3: Details of Monte Carlo Sampling
> We kindly refer the reviewer to our response to Q1 of Reviewer rAmJ for more details.
>
> # Q4: Details of training.
> For the detailed architecture of STI Estimation Head, please refer to SupMat.
>
> For the training pipeline, our PST is trained in two stages using the same optimizer configuration, with a same batch size of 128 and 100 epochs for each stage.
>
> In the first stage, we train a teacher STI Estimation Head independently using Monte Carlo (MC) sampling to approximate the STI values based on loss function in Eq. (6). The teacher STI Estimation Head learns to match these MC–estimated interaction distributions and serves as an efficient approximation module. In the second stage, both joint-wise and segment-wise modules incorporate a student STI Estimation Head to learn which fine-grained token pairs are critical for matching. The trained teacher STI Estimation Head provides token-level supervision in the form of an STI value distribution. This interaction-based signal guides the model to focus on essential correspondences, enabling fine-grained alignment and improving retrieval performance.
> We will revise the manuscript to explicitly clarify this two-stage training procedure and the distinction between the STI Estimation Head and the retrieval model to improve readability and reproducibility.
>
>
> # Q5: Adaptation of PST to other motion representations.
>
> PST is inherently representation-agnostic, as it operates on tokenized motion features rather than relying on MotionPatch. We have already validated this generality in our ablation study by replacing MotionPatch with the Guo feature, where PST still achieves reasonable performance, although slightly degraded due to the weaker structural expressiveness. Furthermore, we also evaluate PST using raw joint coordinates. The results are summarized below. This indicates that PST does not depend on MotionPatch, but benefits from more structured representations.
>
> |Representation|(Text→Motion)R@1↑|R@5↑|R@10↑|MedR↓|(Motion→Text)R@1↑|R@5↑|R@10↑|MedR↓|
> |-|-|-|-|-|-|-|-|-|
> |Raw Joint Coordinates|8.56|28.18|41.87|17.00|10.11|28.54|35.94|16.00|
> |Guo Feature|8.11|27.15|39.89|18.00|9.64|26.31|34.55|17.00|
> |**PST**|**12.45**|**33.65**|**48.22**|**10.00**|**13.59**|**31.88**|**42.68**|**15.00**|

---

> > ### Author Rebuttal · Reviewer_6aDZ · 2026-04-06
> >
> > The authors have addressed my concerns and thus I maintain my positive score (4) on it.

---

### Decision · Program_Chairs · 2026-04-30

**Decision:**

Accept (spotlight)

**Comment:**

The reviewers agree that the paper addresses an important problem in motion-language retrieval and that its central idea is technically sound and empirically promising. Across the reviews, the main strengths were the clear motivation for moving beyond global alignment, consistent performance gains on standard benchmarks, and useful qualitative visualizations supporting fine-grained correspondence learning. The principal concerns were about whether the pyramidal design and overall novelty were sufficiently distinguished from prior structured alignment approaches, limited experimental breadth, and inadequate clarity on computational cost, Monte Carlo sampling, training procedure, and several implementation choices. The reviewers differed mainly in how serious they viewed these issues: some regarded the framework as conceptually novel and significant, while others saw the novelty as more incremental and the gains as moderate relative to the added complexity. The rebuttal substantially strengthened the paper by clarifying the two-stage training procedure, reporting efficiency analyses, adding ablations on the alignment stages and KNN-DPC, and providing additional comparisons and larger-scale evaluation; reviewers explicitly indicated that their major concerns were resolved.